# A novel 3D visualization method in mice identifies the periportal lamellar complex (PLC) as a key regulator of hepatic ductal and neuronal branching morphogenesis

**Tongtong Xu[1†], Fujun Cao[1†], Ruihan Zhou[1†], Qin Chen[2], Jian Zhong[3], Yulin Wang[1], Chaoxin Xiao[1], Banglei Yin[1], Chong Chen[1*], Chengjian Zhao[1*]**

[1]State Key Laboratory of Biotherapy and Cancer Center, West China Hospital, Sichuan University, and Collaborative Innovation Center for Biotherapy, Chengdu, China; [2]Chengdu Minghong Tiancheng Technology Co., Ltd, Chengdu, China; [3]Peking University Chengdu Academy for Advanced Interdisciplinary Biotechnologies, Chengdu, China

**\*For correspondence:**
chongchen@scu.edu.cn (CC);
chjianzhao@scu.edu.cn (CZ)

[†]These authors contributed equally to this work

## eLife Assessment

This study presents an **important** methodological advance-Liver-CUBIC combined with multicolor metallic nanoparticle perfusion-that enables high-resolution 3D visualization of the liver's complex multi-ductal architecture. The identification of the periportal lamellar complex (PLC) as a novel perivascular structure with distinct cellular composition and low-permeability characteristics is **convincing**, supported by rigorous imaging data. The observed scaffolding role during fibrosis offers intriguing biological insights, though the functional claims would benefit from direct experimental validation.

**Abstract** The liver is a complex organ responsible for multiple functions, including metabolism, energy storage, detoxification, bile secretion, and immune regulation. Its highly organized vascular system plays a crucial role in maintaining functional zonation and tissue homeostasis. Within the liver, the hepatic artery, portal vein, hepatic vein, bile duct, and nerve networks intertwine to form an intricate three-dimensional architecture; however, traditional two-dimensional imaging fails to reveal their true spatial relationships, and current three-dimensional imaging methods remain insufficient to capture fine structural details. To achieve comprehensive visualization of these multi-ductal systems, we established a high-resolution three-dimensional imaging platform that combines multicolor perfusion of metallic compound nanoparticles (MCNPs) with an optimized tissue-clearing protocol (Liver-CUBIC), enabling simultaneous 3D reconstruction of the portal vein, hepatic artery, bile duct, and hepatic vein in mouse livers. Based on these data, we identified and defined a previously unrecognized structure located in the outer layer of the portal vein, termed the periportal lamellar complex (PLC). The PLC encircles the portal vein between the vascular endothelium and the perisinusoidal region, exhibits low-permeability barrier characteristics, and contains a distinctive population of CD34+Sca-1+ endothelial cells. During liver fibrosis, the PLC extends from the portal vein toward the hepatic lobule, forming a structural scaffold that guides bile duct and nerve migration.

## Introduction

The liver performs diverse physiological functions, including nutrient metabolism, detoxification, bile secretion, and immune regulation, all of which depend on its unique microanatomy—the hepatic lobule, the smallest structural and functional unit (*Almazroo et al., 2017*; *Friedman, 2008*; *Kalra et al., 2023*; *Ozougwu, 2017*). To maintain efficient operation across the vast hepatic parenchyma, the liver has evolved a highly complex and densely organized ductal vascular-neuronal network in the body, consisting primarily of the portal vein system, central vein system, hepatic artery system, biliary system, and intrahepatic autonomic nerve network (*Miller et al., 2021*; *Zhang et al., 2022*). These systems form a multicompartmental 3D network that integrates blood transport, bile drainage, and neuroimmune regulation essential for maintaining hepatic homeostasis. Despite their distinct roles, these systems are spatially and functionally intertwined to coordinate metabolism, immunity, and regeneration within each lobule (*Zhang et al., 2022*). While macroscopic vascular patterns and metabolic zonation have been described (*Lv et al., 2021*; *Peeters et al., 2018*), the micro-level interactions among these ductal systems within individual lobules remain poorly understood.

Traditional two-dimensional (2D) histology provides high spatial resolution but cannot reconstruct the true 3D continuity of vascular structures or their relationships across multiple ductal systems (*Hankeova et al., 2021*; *Kim et al., 2015*). Thus, three-dimensional (3D) visualization of multiple intrahepatic networks is essential for understanding liver development, homeostasis, and disease. Conventional imaging modalities such as ultrasound, micro-computed tomography (micro-CT), and magnetic resonance imaging (MRI) offer macroscopic resolution but cannot resolve micrometer-scale vascular details. X-ray phase-contrast computed tomography (PCCT) achieves contrast-agent-free microvascular imaging in rat livers (*Qin et al., 2017*), yet still lacks subcellular resolution. Similarly, the double resin-casting micro computed tomography (DUCT) technique reconstructed 3D portal vein–bile duct networks using dual-color resin casting, but failed to visualize luminal structures smaller than 5 μm (*Hankeova et al., 2021*). Therefore, a high-resolution, multicolor 3D imaging approach is required to elucidate terminal interactions such as portal vein–artery coupling, bile duct–vessel countercurrent exchange, and nerve–vessel communication.

Developing such imaging approaches for the liver remains technically challenging. The hepatic vasculature is highly dense (sinusoids occupy 10–15% of the liver volume) and intricately intertwined, demanding submicron resolution and millimeter-scale penetration depth (*Lv et al., 2021*; *Segovia-Miranda et al., 2019*). Moreover, tissue-specific autofluorescence (e.g., from bilirubin, lipofuscin, collagen, and NADPH/FAD) impairs optical clearing and fluorescence signal quality, yielding low signal-to-noise ratios (SNR) (*Molina et al., 2022*; *Susaki et al., 2014*; *Tainaka et al., 2014*). Finally, the liver's multiple intrahepatic systems—blood vessels, bile ducts, nerves, lymphatics, and immune networks—are functionally interdependent, yet current methods lack efficient 3D multi-target labeling strategies to visualize them simultaneously. Consequently, existing clearing and labeling protocols (such as CLARITY, iDISCO, and conventional immunofluorescence) remain limited in depth, resolution, and SNR, leaving the liver far behind other organs in 3D multicolor imaging (*Liu et al., 2022*; *Molina et al., 2022*).

To address these challenges, we developed a liver-adapted, high-fidelity 3D visualization strategy tailored for the hepatic multi-ductal system. This platform integrates several key innovations: (1) the optimized Liver-CUBIC tissue clearing protocol shortened the clearing duration by 63.89% and enhanced optical transmittance by 20.12%, compared with the original CUBIC protocol; (2) multicolor metal compound nanoparticles (MCNPs), which provide stable, simultaneous labeling of the portal vein, hepatic artery, bile duct, and central vein, and support multimodal imaging at subcellular resolution; and (3) a multimodal immunolabeling system compatible with 3,3'-diaminobenzidine (DAB) and tyramide signal amplification (TSA)-based multiplex staining, allowing high signal-to-noise 3D mapping of vascular and cellular structures.

Using this platform, we achieved liver 3D visualization at single-cell resolution and discovered a previously unrecognized structure, the periportal lamellar complex (PLC). The PLC is periodically distributed along the portal vein axis and closely associated with terminal bile duct branches and autonomic nerve plexuses, forming a specialized interaction unit. Single-cell transcriptomic and multiplex 3D immunostaining analyses revealed that the PLC exhibits distinct localization, morphology, and gene expression signatures, suggesting it may play a crucial role in hepatic microanatomical coordination.

In summary, this liver-specific multicolor 3D visualization system enables comprehensive mapping of hepatic ductal–vascular interactions from the organ to the single-cell level. It not only demonstrates the power of the platform in resolving complex hepatic microstructures but also establishes a new technical foundation for investigating hepatic physiology and pathological remodeling.

## Results

### $H_2O_2$-enhanced CUBIC clearing with multicolor nanoparticles enables high-resolution 3D mapping of the hepatic ductal–vascular network

The CUBIC (Clear, Unobstructed Brain/Body Imaging Cocktails and Computational analysis) technique has been widely applied in the tissue clearing and 3D imaging of organs such as the brain, kidney, and heart due to its excellent compatibility with various fluorescent dyes and immunolabeling protocols (*Susaki et al., 2014*; *Tainaka et al., 2014*). However, the application of CUBIC in liver tissue remains challenging because of the liver's high heme content and densely packed cellular architecture, which leads to pronounced light scattering and markedly limits the clearing efficacy. Furthermore, the conventional CUBIC protocol requires up to 9 days for liver tissue processing, severely restricting experimental throughput (*Figure 1A*, red schematic). To address these limitations, Molina et al. incorporated a hydrogen peroxide ($H_2O_2$) bleaching step into the CLARITY protocol, improving liver transparency but failing to reduce the processing time (*Molina et al., 2022*). In this study, we developed an optimized clearing strategy termed Liver-CUBIC, featuring two key modifications: (1) integration of an $H_2O_2$-CUBIC hybrid system, incorporating an $H_2O_2$ bleaching step into the CUBIC workflow. This technical improvement effectively removes pigments such as heme and lipofuscin through oxidative reactions, while partially clearing lipid components to reduce light scattering; (2) optimization of the urea concentration, increased from 25% to 40%, enhanced protein denaturation, thereby reducing light scattering and improving tissue permeability, which promoted more uniform reagent distribution. As a result, the optimized Liver-CUBIC protocol significantly shortened the liver clearing time (*Figure 1—figure supplement 1A*). Specifically, initial PBS/PFA perfusion efficiently removed residual heme; subsequent $H_2O_2$ bleaching eliminated pigment deposits; and final 40% urea treatment achieved refractive index homogenization (*Figure 1—figure supplement 1A*). Using this optimized protocol, the total clearing time was reduced from 9 days with conventional CUBIC (*Figure 1A*, red schematic) to 3.25 days with 40% urea + $H_2O_2$ treatment (*Figure 1A*, green schematic), corresponding to a 63.89% reduction in clearing time. Importantly, Liver-CUBIC treatment did not induce significant tissue expansion (*Figure 1B–D*). In addition, quantitative transmittance measurements in 1-mm-thick cleared tissue slices showed an average increase of 20.12% (p<0.0001; 95% CI: 19.14–21.09; *Figure 1E*). Additionally, in 1-mm-thick liver sections, the clearing time was reduced to just 20 min (*Figure 1—figure supplement 1B*), while real-time imaging demonstrated complete clearing of 200-μm-thick slices within 60 s (*Figure 1—figure supplement 1C*, and *Figure 1—video 1*), fully meeting the demands for rapid live imaging.

MCNPs exhibit excellent brightness, tunable multicolor properties, and strong resistance to $H_2O_2$ bleaching, maintaining color stability even after Liver-CUBIC clearing. Utilizing four distinct MCNP colors—pink, green, black, and yellow—we developed a four-channel vascular labeling strategy (*Figures 1D*). The hepatic artery was perfused with yellow MCNPs via the left ventricle, bile ducts were retrogradely injected with green MCNPs through the extrahepatic duct, portal veins were directly filled with pink MCNPs, and central veins were labeled using black MCNPs via the inferior vena cava. Using this method, simultaneous four-color labeling of the intrahepatic ductal–vascular system can be achieved (*Figure 1—figure supplement 1E–G*), while single- or dual-color labeling allows for higher-resolution visualization of fine microscale structures. High-resolution 3D imaging revealed that MCNP-Green-labeled large intrahepatic bile ducts exhibit a one-to-one parallel topology with adjacent portal vein branches. At the periphery, the terminal biliary network forms a classic three-dimensional plexus, characterized by helical configurations and polygonal extensions projecting into the hepatic lobules (*Figure 1G*). Meanwhile, MCNP-Pink-labeled portal veins displayed a typical tree-like branching pattern in the central regions of hepatic lobes, but in the marginal areas, they formed an unusually dense, radial micro-branching pattern (*Figure 1H*). Notably, the terminal micro-branches of the portal vein appeared not only around secondary branches but also intermittently along the main trunks, challenging the conventional notion of strictly hierarchical vascular bifurcation.

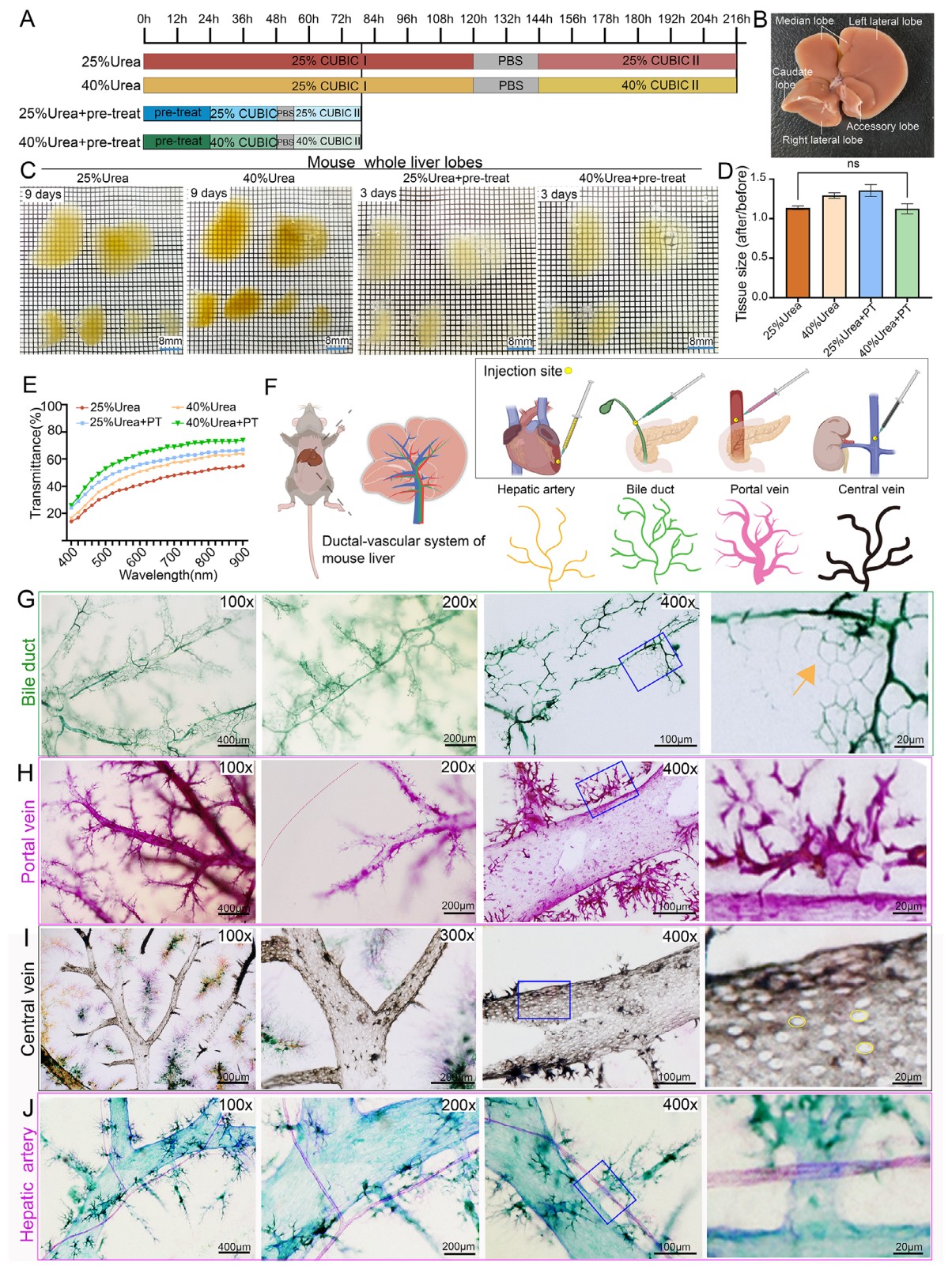

**Figure 1.** H₂O₂-enhanced CUBIC clearing with multicolor nanoparticles enables high-resolution 3D mapping of the hepatic ductal–vascular network. (**A**) Comparison of processing times for four different liver clearing protocols: conventional CUBIC (25% urea, red), high-concentration urea (40% urea, yellow), oxidation treatment alone (25% urea +4.5% H₂O₂, blue), and the optimized Liver-CUBIC (40% urea +4.5% H₂O₂, green). The optimized Liver-CUBIC protocol significantly reduced clearing time (n=6 per group). (**B**) Schematic illustration of mouse liver lobes, defining left lateral lobe, left

*Figure 1 continued on next page*

*Figure 1 continued*

medial lobe, right medial lobe, right lateral lobe, caudate lobe, and quadrate lobe according to reference (*Zhang and DeBosch, 2022*). (**C**) Brightfield images of whole mouse livers (8–9 weeks old, male) after processing with the four clearing protocols. Grid size: 1.6 mm × 1.6 mm. Scale bar: 8 mm. (**D**) Quantitative analysis of tissue volume changes following each clearing protocol (n=6 per group). No statistically significant difference was observed between the 25% urea group and the 40% urea + PT group. Statistical test: unpaired two-tailed *t*-test. PT: pre-treatment. (**E**) Transmission spectra (400–900 nm) of 1-mm-thick mouse liver samples after each clearing protocol (n=5 per group). (**F**) Schematic diagram of the four-channel ductal–vascular labeling strategy: yellow MCNPs injected via the left ventricle to label hepatic arteries, green MCNPs via retrograde common bile duct injection to label biliary ducts, pink MCNPs via the portal vein trunk to label portal veins, and black MCNPs via the inferior vena cava to label hepatic veins. Note that MCNP colors are used for visualization purposes and do not indicate a fixed dye–structure correspondence. (**G**) Three-dimensional fine structures of the biliary tree in the central and peripheral regions of the mouse liver, labeled with MCNP-Green. Images are shown at magnifications of 100×, 200×, and 400×. The rightmost panel presents a high-magnification view of the area outlined by the blue box. Scale bars: 400 μm, 200 μm, 100 μm, and 20 μm. The arrow indicates the terminal ductal structures with a polygonal shape. (**H**) Three-dimensional reconstruction of the portal venous system labeled with MCNP-Pink, showing branching features in central and peripheral zones. Images captured at 100×, 200×, and 400× magnifications; the rightmost image is a higher-magnification view of the blue-boxed region. Scale bars: 400 μm, 200 μm, 100 μm, and 20 μm. (**I**) Three-dimensional structural details of the central vein labeled with MCNP-Black. MCNP-Yellow, MCNP-Pink, and MCNP-Green were used to label the portal vein, hepatic artery, and bile duct, respectively. Images captured at 100×, 300×, and 400× magnifications; the rightmost image is a higher-magnification view of the blue-boxed region. Scale bars: 400 μm, 200 μm, 100 μm, and 20 μm. (**J**) Three-dimensional structural details of the hepatic artery labeled with MCNP-Pink. Images captured at 100×, 200×, and 400× magnifications; the rightmost image is a higher-magnification view of the blue-boxed region. Scale bars: 400 μm, 200 μm, 100 μm, and 20 μm.

The online version of this article includes the following video, source data, and figure supplement(s) for figure 1:

**Source data 1.** Quantitative analysis of tissue volume changes under different clearing protocols (n=6 per group).

**Source data 2.** Transmission spectra (400–900 nm) of 1-mm-thick mouse liver samples following each clearing protocol (n=5 per group).

**Figure supplement 1.** Establishment of a method for simultaneous three-dimensional visualization of the mouse hepatic vascular system.

**Figure 1—video 1.** Real-time imaging of 200-μm-thick liver slices showing complete clearing within 60 seconds.

https://elifesciences.org/articles/108669/figures#fig1video1

For MCNP-Black-labeled central veins, imaging revealed sparse, circular or oval fenestrae measuring 5–10 μm in diameter distributed across the surface of the central vein trunks (*Figure 1I*). Previous scanning electron microscopy studies have described similar structures, which are thought to facilitate the return of hepatocyte-synthesized metabolic products (such as lipoproteins and cytokines) into the circulation, assist immune cell trafficking from the liver to the systemic circulation, and help maintain hepatic sinusoid–interstitium osmotic pressure balance (*Mak and Shin, 2021*). Additionally, dual-label imaging using MCNP-Pink and MCNP-Green demonstrated that the hepatic artery runs parallel to the portal vein trunk, with the hepatic artery displaying a notably smaller diameter compared to its adjacent portal vein. At the periphery, hepatic arterial branches formed a three-dimensional entwined network that closely contacted the portal vein micro-branches (*Figure 1J*).

In summary, combining Liver-CUBIC with MCNPs enabled highly efficient, simultaneous mapping of intrahepatic bile ducts, portal veins, central veins, and hepatic arteries in mice at cellular resolution, revealing their spatial organization and topological relationships.

## The PLC serves as a low-permeability gateway bridging portal veins to hepatic lobules

In this study, scanning electron microscopy revealed that MCNP dye is a nanoparticle-based labeling agent with a diameter of approximately 150–200 nm (*Figure 2A*). Our experimental results showed that MCNPs were primarily distributed within the lumens of blood vessels and bile ducts through physical perfusion, without exhibiting specific molecular binding capacity. Under our perfusion conditions, the dye signals remained confined to the luminal regions, clearly delineating the morphology of vascular and biliary structures without diffusing into the surrounding parenchyma (*Figure 2—figure supplement 1A–H*). However, due to the increased imaging depth of liver tissue after optical clearing, nuclear stains such as DAPI often generate highly dense fluorescence signals that obscure fine vascular and perivascular structures (*Figure 2—figure supplement 1I and J*). Thus, our imaging analyses primarily focused on the high-resolution visualization of the spatial architecture of the vascular and biliary systems.

We employed a dual-channel vascular labeling strategy to simultaneously visualize the portal vein and hepatic artery systems in the mouse liver. Specifically, MCNP-Green was employed to label the

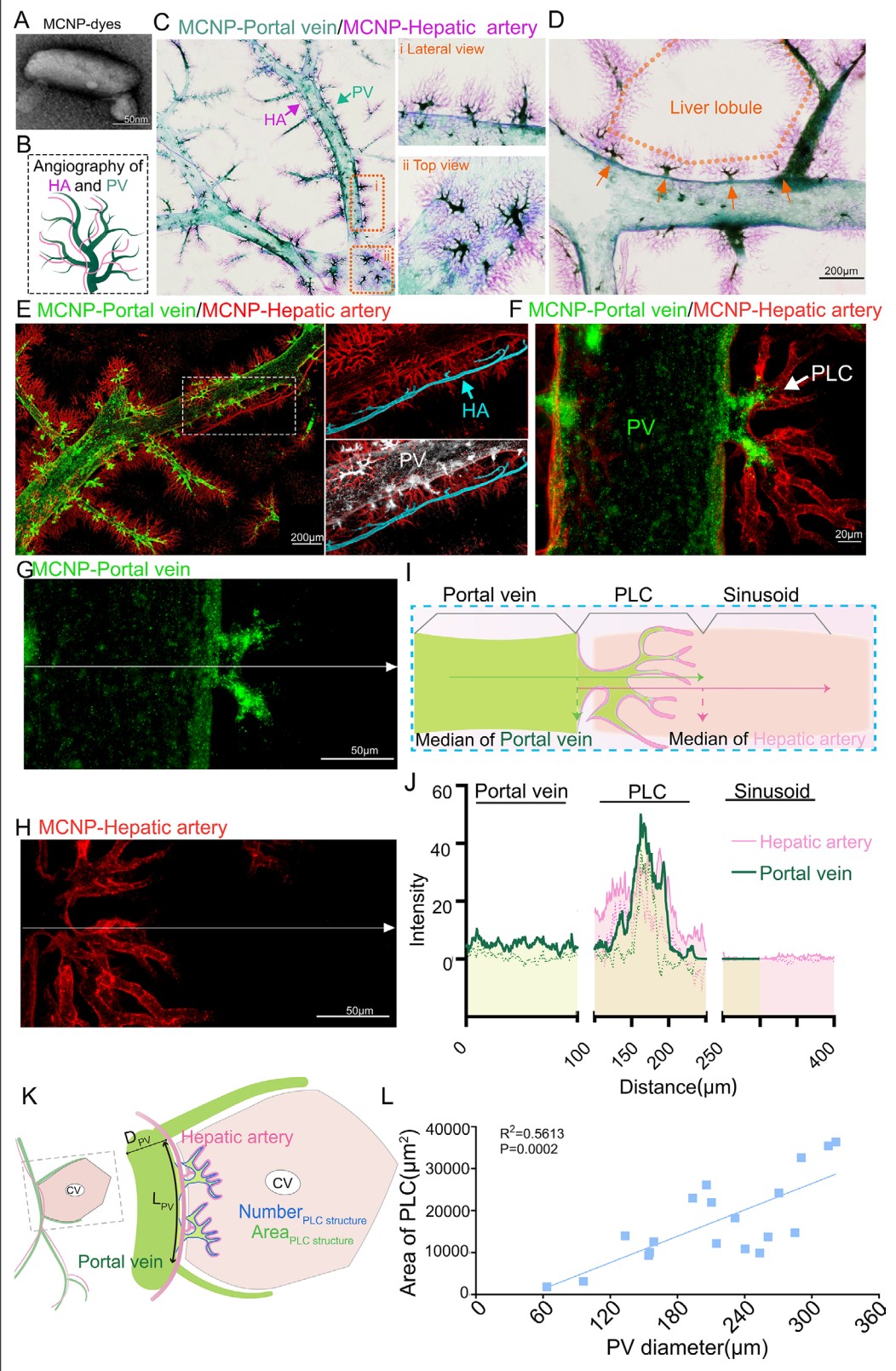

**Figure 2.** The periportal lamellar complex (PLC) serves as a low-permeability gateway bridging portal veins to hepatic lobules. (**A**) Scanning electron microscopy revealed that the metal compound nanoparticles (MCNP) dyes consisted of aggregates of particles approximately 150–200 nm in length. (**B**) Dual-channel vascular typing labeling scheme (Dual-channel NanoFluor). hepatic arteries labeled with pink fluorescent nanoparticles portal

*Figure 2 continued on next page*

*Figure 2 continued*

veins labeled with green fluorescent nanoparticles. (**C**) EDOF imaging (tissue thickness: 200 μm) reveals fine three-dimensional structural details of the hepatic artery and portal vein. The right panel shows magnified lateral and top views of the orange-boxed region, highlighting dense terminal branches and the close spatial proximity between the two vascular systems. Green arrows indicate portal veins, and pink arrows indicate hepatic arteries. (**D**) Magnified image showing periodic alignment of PLC structures (orange arrows) along the adventitial layer of the portal vein trunk (green) in the direction of vascular flow. Orange dashed lines delineate the boundaries of a classical hepatic lobule. Scale bar: 200 μm. (**E**) Three-dimensional confocal imaging of hepatic arteries (pink dye, excitation/emission: 561/640 nm) and portal veins (green dye, excitation/emission: 495/519 nm). The right panel presents a high-resolution view with hepatic arteries rendered in blue and the portal vein surface filled in white. Scale bar: 200 μm. (**F**) High-magnification confocal imaging further depicting micro-branches of portal veins and hepatic arteries, with terminal branches intertwining in a coiled distribution. arrows indicate points of interaction between PLC structures and hepatic arteries. Scale bar: 20 μm. (**G, H**) Single-channel confocal images showing the distribution of MCNP-Green-labeled portal veins (**G**) and MCNP-Pink-labeled hepatic arteries (**H**) within PLC regions. White arrows indicate the paths of fluorescence intensity profile measurements, with arrowheads denoting the direction of line scans. Scale bar: 50 μm. (**I**) Schematic illustration of fluorescence intensity profile measurements across PLC structures. The midpoint of the portal vein (green) intensity profile corresponds to the junction between the PLC and the outer wall of the portal vein, while the midpoint of the hepatic artery (pink) intensity profile aligns with the terminal edge of the PLC adjacent to liver sinusoids. Arrows indicate scan directions. (**J**) Fluorescence intensity profile plots. The X-axis represents the scan distance (0–400 μm), and the Y-axis represents fluorescence intensity. Both portal vein (green) and hepatic artery (pink) signals showed significant increases within the PLC region. The regions were defined as follows: 0–100 μm, portal vein region; 100–250 μm, PLC region; 250–400 μm, liver sinusoid region (n=5 per group). (**K**) Analytical workflow for characterizing PLC structures: the primary portal vein trunk carrying PLC structures was selected as the reference axis for hepatic lobule boundaries, combined with its two adjacent secondary branches to define a classical hexagonal lobule computational unit. The diameter of the primary portal vein trunk (dPV) and the area of PLC structures were quantified using the extended depth-of-field imaging system. (**L**) Distribution of PLC areas along primary portal vein trunks with diameters ranging from 63.45 to 321.42 μm. Each value represents the PLC area associated with a portal vein of corresponding diameter (n=19).

The online version of this article includes the following source data and figure supplement(s) for figure 2:

**Source data 1.** Fluorescence intensity profiles showing increased signals for the portal vein and hepatic artery within the PLC region.

**Source data 2.** Distribution of PLC areas along primary portal vein trunks with diameters ranging from 63.45 to 321.42 μm.

**Figure supplement 1.** Analysis of periportal lamellar complex (PLC) distribution along portal veins by length, diameter, and area.

**Figure supplement 1—source data 1.** Distribution of PLC structures along portal vein trunks with lengths ranging from 269.85 μm to 1513.67 μm.

**Figure supplement 1—source data 2.** Distribution of total PLC area along portal vein trunks with lengths ranging from 269.85 μm to 1513.67 μm.

**Figure supplement 1—source data 3.** Distribution of the number of PLC structures along portal vein trunks with diameters ranging from 63.45 μm to 321.42 μm.

portal venous system by injection from the portal vein, while MCNP-Pink marked the hepatic arterial system by injection via the left ventricular (*Figure 2B*).

Extended depth-of-field (EDOF) imaging revealed that MCNP-Green specifically labels a distinct vascular structural unit located in the adventitial layer of the portal vein trunk, which we designate as the PLC (*Figure 2C and D*, orange arrows). Morphologically, this structure consists of fine vascular branches distributed along the surface of the portal vein trunk. Its base is anchored to the portal vein, and it radiates outward in a lobular pattern centered on the portal vein, giving rise to multiple terminal branches that directly connect to the liver sinusoids (*Figure 2C*). The PLC structures align in a serial arrangement along the portal vein trunk (*Figure 2D*).

Further validation by confocal fluorescence imaging confirmed that following portal vein perfusion with MCNP-Green, fluorescence signals are highly enriched within the PLC. Quantitative analysis showed that the fluorescence intensity within the PLC was significantly higher than that observed in the portal vein lumen and liver sinusoid regions, with minimal dye diffusion at the PLC periphery,

indicating predominant localization within the main PLC structure (*Figure 2E–J*). In contrast, MCNP-Pink signals, introduced via hepatic artery perfusion, predominantly localized within the PLC and its margins without crossing into adjacent PLC units. Due to differences in perfusion pathways, the pink dye demonstrated a greater tendency to extend from the PLC towards the lobular parenchyma compared to the green dye. This phenomenon likely reflects the relatively low permeability of the PLC region: green nanoparticles delivered through the portal vein are largely retained within the low-permeability PLC core, with limited peripheral diffusion, whereas the arterial pink dye, influenced by the pressure and flow characteristics of the hepatic artery system, effectively labels the peripheral PLC structures (*Figure 2G–J*).

To further characterize PLC distribution, we used the EDOF imaging system to quantitatively analyze the distribution of PLC structures in relation to portal vein diameter and anatomical location. By defining classical hexagonal hepatic lobule units through the primary portal vein trunk containing PLC structures and its two adjacent secondary branches (*Figure 2K*), quantitative analysis revealed that the length of the primary portal vein trunk ranged from 269.85 μm to 1513.67 μm (n=19), with a median length of 769.89 μm (*IQR*: 593.67–987.77 μm), the number of PLC structures per trunk ranged from 1 to 4, with 68.42% (13/19) of portal veins containing two PLCs (*Figure 2—figure supplement 1K*). Notably, neither the number nor the area of PLCs showed significant correlation with the length of the portal vein trunk (*Figure 2—figure supplement 1K and L*), indicating that morphological parameters alone (such as trunk length) may be insufficient to define hepatic lobular units, especially given the substantial differences in blood transport capacity among portal veins of varying diameters. The diameters of the primary portal veins bearing PLC structures ranged from 63.45 μm to 321.42 μm, and PLC area showed a significant positive correlation with portal vein diameter (*Figure 2L*), whereas PLC number did not correlate with diameter (*Figure 2—figure supplement 1M*). This diameter–area relationship may reflect a functional adaptation of the portal venous system in metabolically active liver regions: larger-diameter portal veins accommodate higher blood flow and consequently experience greater shear stress on the vessel wall. The formation of PLC structures in the adventitial layer may participate in local blood flow regulation, maintenance of microenvironmental homeostasis.

These findings suggest that the PLC is a morphologically and spatially distinct vascular structure surrounding the portal vein, which likely serves as a key organizational node coordinating the spatial relationships among the portal vein, hepatic artery, and liver sinusoids. Thus, the PLC represents an important structural element within the interactive vascular network of the mouse liver.

## Spatial juxtaposition of the periportal lamellar complex with canals of Hering at the portal venous interface

To further investigate the potential relationship between the PLC and bile delivery regulation within the hepatic lobule, we employed a dual-channel vascular-biliary labeling strategy to simultaneously visualize the mouse liver portal vein and bile duct systems.

We combined MCNP-Pink labeling of the portal vein and its associated PLC structures with three-dimensional DAB immunohistochemical staining for CK19 to visualize the bile duct epithelial architecture. EDOF imaging revealed a consistent 1:1 anatomical association between the primary portal vein trunk (diameter 280±32 μm) and the first-order bile duct (diameter 69 ± 8 μm) (*Figure 3A*, *Figure 3—figure supplement 1*). Multiview imaging showed that second-order bile ducts are distributed along the portal vein trunk and branch directionally toward the PLC. The terminal portions, namely the canals of Hering, form the distal bile duct network responsible for bile collection and are located at the interface between hepatic lobules, rather than being evenly distributed around the portal vein. These terminal ducts are specifically concentrated within the PLC surrounding the portal vein, displaying marked spatial colocalization (*Figure 3A*). This finding suggests a potential anatomical spatial association between the terminal canals of Hering and the periportal PLC structures. Furthermore, based on MCNP-Green labeling of the biliary tract, combined with 3D DAB-CK19 staining, both EDOF imaging and 3D confocal fluorescence imaging consistently confirmed that the terminal regions of the biliary network—responsible for collecting bile from the hepatic parenchyma—are primarily located adjacent to the PLC (*Figure 3B and C*).

To delineate the microscopic architecture underlying bile flow from the hepatic parenchyma into the bile duct network, we employed three-dimensional TSA-based multiplex immunofluorescence to simultaneously label the tight junction protein ZO-1, the hepatocyte nuclear marker HNF4A, and

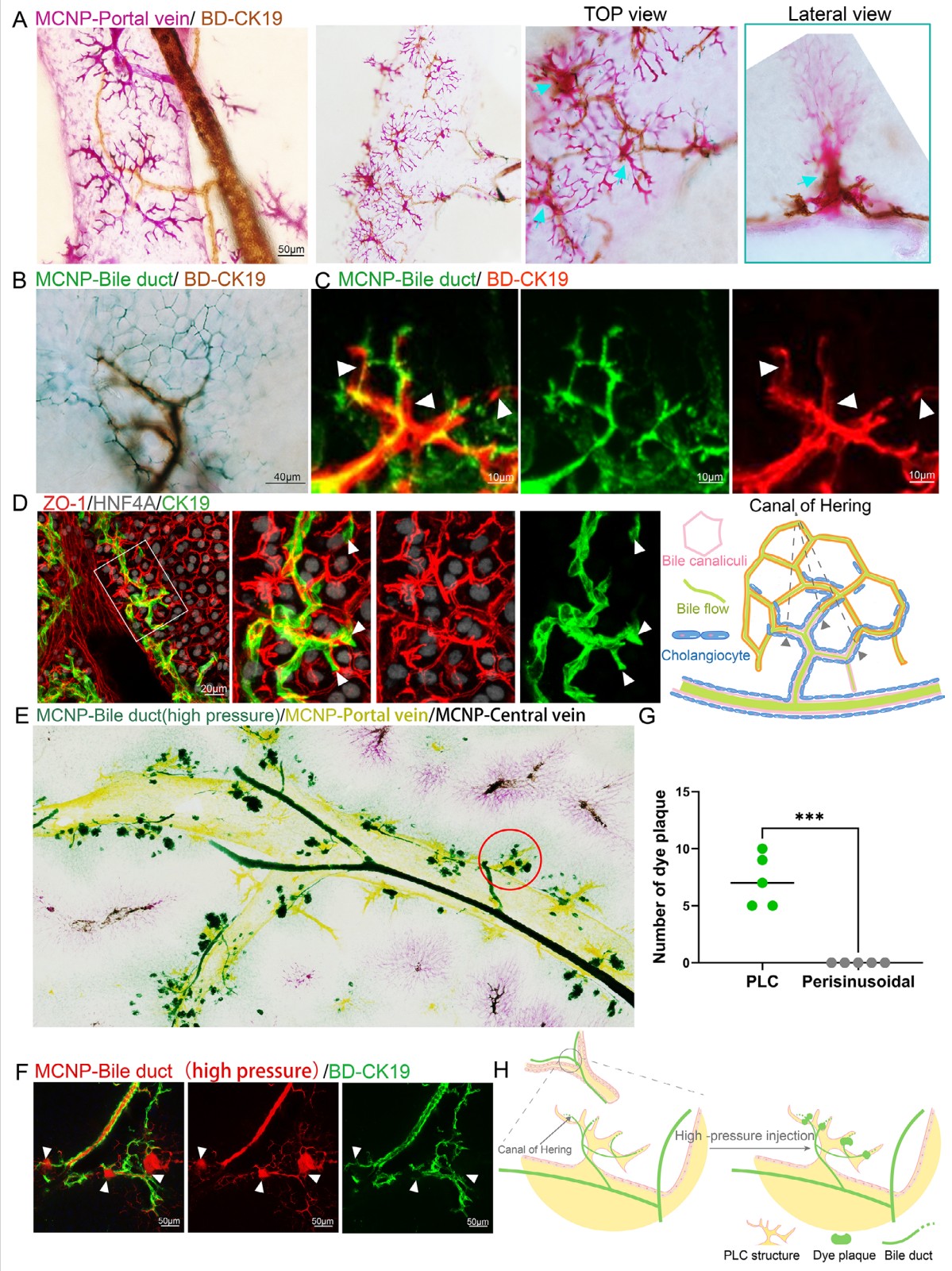

**Figure 3.** Spatial juxtaposition of the periportal lamellar complex with canals of Hering at the portal venous interface. (**A**) MCNP-Pink labeling of portal veins combined with three-dimensional DAB immunohistochemistry for CK19 (brown) to visualize bile duct epithelial cells. Top and lateral views highlight the PLC–bile duct interaction sites (cyan arrows). Scale bar: 50 μm. (**B**) MCNP-Green labeling of bile ducts combined with three-dimensional DAB immunohistochemistry for CK19 (brown). Scale bar: 40 μm. (**C**) MCNP-Green labeling of bile ducts combined with three-dimensional TSA

*Figure 3 continued on next page*

*Figure 3 continued*

immunofluorescence for CK19 (red), displaying detailed structures at the interface between dye-labeled ducts and immunostained bile duct terminals. The arrows indicate the terminal structures of the bile ducts. Scale bar: 10 μm. (**D**) Three-dimensional TSA multiplex immunofluorescence staining for ZO-1 (red, marking bile canaliculi networks), HNF4α (gray, marking hepatocyte nuclei), and CK19 (green, marking bile ducts). Scale bar: 20 μm. The right panel illustrates the spatial relationship between ZO-1-labeled bile canaliculi and CK19-labeled bile duct terminals, with arrows indicating the terminal positions of bile ducts. (**E**) EDOF imaging of the whole liver showing high-pressure perfusion of green fluorescent nanoparticles into the bile duct and yellow nanoparticles labeling the portal vein. Pink signal (arrow) indicates hepatic artery labeling, which may appear slightly displaced toward the central vein due to minor misalignment during perfusion. Red circles indicate sites of green dye leakage localized to the PLC regions. (**F**) High-pressure retrograde perfusion of red fluorescent nanoparticles into the bile duct, combined with three-dimensional TSA immunofluorescence for CK19 (green). Arrows indicate sites of dye leakage at the PLC region. Scale bar: 50 μm. (**G**) Statistical analysis of the number of dye leakage foci shown in the left panel (**H**), comparing the PLC with the perisinusoidal regions adjacent to the portal vein. *** indicates a highly significant difference between groups ($p<0.0001$). (**H**) Schematic diagram illustrating the leakage sites of bile duct-perfused dye following high-pressure injection. Green dashed boxes represent the positions of free bile duct terminal epithelial cells at leakage sites.

The online version of this article includes the following source data and figure supplement(s) for figure 3:

**Source data 1.** Statistical analysis of dye leakage foci comparing PLC and perisinusoidal regions adjacent to the portal vein ($P < 0.0001$).

**Figure supplement 1.** Interactions between PLC structures and terminal biliary tree in the mouse liver.

(**A**) Portal veins labeled with pink metallic nanoparticle dye combined with three-dimensional DAB immunohistochemistry (CK19[+], brown) marking biliary epithelial cells. A larger view shows the accompanying distribution of bile ducts along portal veins. (**B**) Portal veins labeled with pink nanoparticle dye and bile ducts labeled with green nanoparticle dye, combined with three-dimensional DAB immunohistochemistry (CK19[+], brown). The boxed region highlights a PLC structure adjacent to terminal bile duct branches. (**C**) Whole-lobe overview of the liver captured by EDOF imaging, showing high-pressure perfusion of green dye for bile ducts and yellow dye for portal veins, visualizing the spatial relationship between biliary and vascular systems.

the bile duct epithelial marker CK19 (*Figure 3D*). Three-dimensional confocal imaging demonstrated that bile canaliculi on the hepatocyte surface formed polygonal networks demarcated by ZO-1, while terminal bile duct epithelial cells also organized into luminal structures via ZO-1, arranged in either single- or multicellular rings, thereby establishing an interface between the canalicular network and the bile duct system. However, detailed morphological analysis suggested that this connection does not occur directly between the bile canaliculi and the terminal bile duct epithelial cells. Instead, bile duct epithelial cells at the terminal ducts extended partially along the canalicular network without directly participating in the formation of the bile duct lumen (*Figure 3D*). This finding indicates that these cells may remain in an immature state or possess alternative stem/progenitor cell-like properties (*Banales et al., 2019*; *de Jong et al., 2021*).

To further verify the precise interface between the terminal bile ducts and the bile canalicular network, we performed retrograde perfusion of MCNP-Green dye through the common bile duct while progressively increasing the perfusion pressure. When the dye pressure exceeded a defined threshold, dye leakage was observed (*Figure 3E–H*). Quantitative analysis of ultra-deep-focus images demonstrated that the dye leakage sites were primarily located adjacent to the PLC structures, rather than in the perisinusoidal regions surrounding the portal vein (*Figure 3G*).

This observation further supports our earlier hypothesis: first-order bile ducts distribute along the portal vein trunk, secondary bile ducts branch directionally toward the PLC regions, and terminal bile duct branches converge spatially adjacent to the PLC, forming a reproducible periportal spatial arrangement. Together, these findings indicate that the PLC defines a conserved anatomical micro-environment at the portal region that is spatially associated with the organization and terminal positioning of the biliary network within the hepatic lobule.

## Single-cell transcriptomics identifies CD34[+]Sca-1[+] as a unique endothelial signature of the periportal lamellar complex with hematopoietic niche potential

To investigate the molecular characteristics and potential functional roles of endothelial cells (ECs) within the PLC, we re-analyzed publicly available single-cell RNA sequencing (scRNA-seq) data of liver endothelial cells isolated by fluorescence-activated cell sorting (FACS) from three healthy adult mouse livers (*Su et al., 2021*). Unsupervised clustering identified a total of 10 distinct cell clusters (*Figure 4A and B*, *Figure 4—source data 1*). Several clusters contained contaminating non-endothelial populations expressing marker genes for hepatocytes, T cells, cholangiocytes, and macrophages (*Figure 4A and B*), which were excluded from subsequent analyses. After removing these contaminating clusters,

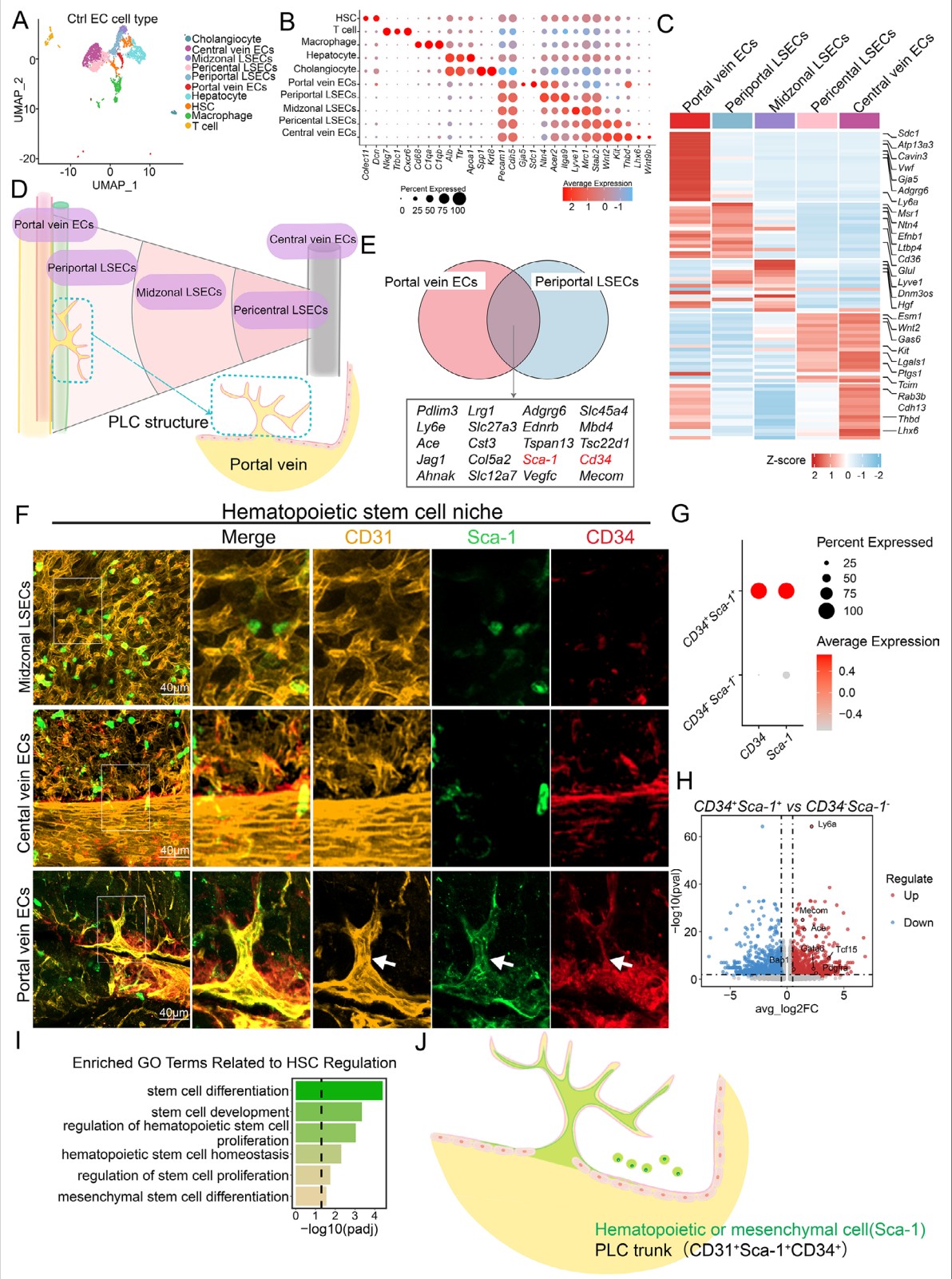

**Figure 4.** Single-cell transcriptomics identifies CD34⁺Sca-1⁺ as a unique endothelial signature of the periportal lamellar complex (PLC) with hematopoietic niche potential. (**A**) UMAP projection of single-cell transcriptomes of liver endothelial cells from normal adult mice reveals 10 distinct cellular clusters. Each dot represents one cell. (**B**) Heatmap showing expression profiles of representative genes across the 10 clusters, including hepatic stellate cells, T cells, macrophages, hepatocytes, cholangiocytes, portal vein ECs, periportal LSECs, midzonal LSECs, pericentral LSECs, and central vein

*Figure 4 continued*

ECs. (**C**) Representative cluster-specific markers for five major liver endothelial subpopulations. (**D**) Spatial schematic illustrating the anatomical position of the PLC, located exclusively between the portal vein ECs and periportal LSECs. (**E**) Venn diagram showing overlapping top 20 highly expressed genes between portal vein ECs and periportal LSECs. (**F**) Multiplex immunofluorescence images showing CD31 (yellow), Sca-1 (green), and CD34 (red) expression in distinct hepatic endothelial zones, including portal vein ECs, midzonal LSECs, and central vein ECs. The PLC structure is demarcated by dashed white arrows. Scale bar: 40 μm. (**G**) Classification of $CD34^+Sca-1^+$ double-positive versus double-negative endothelial subpopulations. (**H**) Volcano plot illustrating differentially expressed hematopoietic-associated genes in $CD34^+Sca-1^+$ cells. The x-axis shows $\log_2$ fold-change ($\log_2FC$); the y-axis shows $-\log_{10}$ adjusted p value. Significantly upregulated genes are located in the upper right quadrant. (**I**) Gene Ontology enrichment analysis of genes upregulated in $CD34^+Sca-1^+$ ECs, highlighting functional categories such as stem cell development, differentiation, and stem cell maintenance. (**J**) The schematic illustration depicts the spatial localization of Sca-1, a marker for mesenchymal or hematopoietic stem cells, and $CD34^+Sca-1^+CD31^+$ endothelial cells as the trunk of the PLC.

The online version of this article includes the following source data and figure supplement(s) for figure 4:

**Source data 1.** Source data for *Figure 4*: Re-analysis of single-cell RNA-seq data from *Su et al., 2021*.

**Figure supplement 1.** Single-cell transcriptomic and immunofluorescence analysis of endothelial cell subpopulations in the mouse liver.

we classified the remaining liver ECs into five subpopulations based on previously reported endothelial cell cluster-specific marker genes (*Halpern et al., 2018*; *Inverso et al., 2021*), the remaining liver ECs were classified into five subpopulations: portal vein ECs, periportal LSECs, midzonal LSECs, pericentral LSECs, and central vein ECs (*Figure 4C*, *Figure 4—figure supplement 1A, B*).

The PLC structures are anatomically located within the transitional zone between portal vein ECs and periportal LSECs (*Figure 4D*). Most characteristic genes within this region exhibited a continuous gradient expression pattern rather than a distinct binary classification. Based on this spatial continuum, we hypothesized that PLC endothelial cells might display a hybrid transcriptional signature, co-expressing marker genes from both adjacent populations (*Figure 4C and D*). Venn diagram intersection analysis combined with differential gene expression screening identified a set of candidate genes potentially specific to PLC endothelial cells (*Figure 4E*). TSA staining confirmed that CD31, a pan-endothelial marker, reliably delineated the spatial boundaries of portal vein ECs, PLC endothelium, and periportal LSECs (*Figure 4F*). Notably, CD34 and Sca-1 (Ly6a) showed continuous co-expression within the PLC region surrounding the portal vein, forming a lamellar endothelial structure (*Figure 4F*). Outside the periportal region, including the midlobular zone, scattered CD34- or Sca-1-positive signals were also detected; however, these signals did not organize into a comparable lamellar architecture.

We further examined the spatial distribution of other endothelial-associated markers in two-dimensional liver sections. CD36 specifically labeled periportal LSECs, while PDGFRβ was broadly expressed across multiple intrahepatic endothelial subtypes (*Figure 4—figure supplement 1C*). Comparative analyses showed that the combined expression of CD34 and Sca-1 was highly enriched in regions associated with the PLC. Based on this, we subsetted $CD34^+Sca-1^+$ endothelial populations from the total liver EC pool (*Figure 4G*). Projection mapping analysis showed that *CD34* and *Sca-1* were primarily distributed within portal vein ECs, periportal LSECs, and midzonal LSECs (*Figure 4—figure supplement 1D*). Given that CD34 and Sca-1 are classical markers of mesenchymal stem cells and intrahepatic hematopoietic stem cells (HSCs), these observations suggest potential stem/progenitor-like properties of PLC-associated endothelial cells (*Agrawal et al., 2024*; *Morcos et al., 2017*; *Petersen et al., 2003*; *Suskind and Muench, 2004*; *Taniguchi et al., 1996*).

Further analysis revealed that the $CD34^+Sca-1^+$ endothelial population marking the PLC structures exhibited significantly upregulated expression of HSC-associated genes (*Figure 4H*). *Mecom*, a critical transcriptional regulator essential for HSC self-renewal and highly expressed in human and approximately 60% of murine long-term HSCs (*Christodoulou et al., 2020*; *Zhang et al., 2011*), was prominently enriched within this population. Additionally, *PDGFRα*, a classical marker for mesenchymal stem cell (MSC) isolation and a potential label for mesenchymal progenitor cells that promote hepatocyte differentiation via direct contact and growth factor secretion (*Kikuchi and Monga, 2015*; *Owen et al., 2022*), was also highly expressed (*Figure 4H*). Gene Ontology (GO) enrichment analysis further revealed that differentially expressed genes in the $Cd34^+Ly6a^+$ population were significantly enriched in pathways associated with the development, proliferation, and differentiation of multiple stem cell types (including hematopoietic and mesenchymal stem cells), as well as in the maintenance of hematopoietic stem cell homeostasis (*Figure 4I*). These findings suggest that this unique endothelial cell

subset in the periportal region may possess dual regulatory functions in both metabolic and hematopoietic modulation.

Finally, the main trunk of the PLC is composed of CD34$^+$Sca-1$^+$CD31$^+$ endothelial cells (*Figure 4J*). These CD34$^+$Sca-1$^+$ double-positive cells were mainly localized to the basal region of the PLC structure and exhibited molecular features associated with hematopoiesis. Together, these findings suggest that PLC endothelial cells may participate in establishing a local microenvironment related to periportal hematopoietic regulation and may play potential roles in stem cell recruitment and vascular homeostasis.

## CD34$^+$Sca-1$^+$ endothelium in the periportal lamellar complex regulates spatial patterning of intrahepatic bile duct branching during liver fibrosis

Our 3D imaging showed that terminal bile duct branches are spatially colocalized with PLC structures around the main portal vein axis (*Figure 5—figure supplement 1A and B*). Volcano plot re-analysis identified multiple genes in the PLC endothelium associated with bile duct development and functional regulation, including *Jcad*, *Dll4*, and *Hes1* (*Figure 5A*). Gene Ontology enrichment analysis confirmed that differentially expressed genes in the *Cd34$^+$Ly6a$^+$* endothelium of PLC are enriched in pathways regulating epithelial morphogenesis and branching morphogenesis of epithelial tubes; this suggests that PLC-associated periportal endothelial cells may be linked to molecular programs related to bile duct structural organization and periportal microenvironmental homeostasis (*Figure 5B*).

During liver fibrosis progression, proliferation and infiltration of terminal bile duct branches (Canals of Hering) into the hepatic parenchyma represent a core feature of the intrahepatic ductular reaction (DR), critically influencing disease progression, liver regeneration, and carcinogenesis (*Carpino et al., 2018*; *Clouston et al., 2005*; *Strazzabosco and Fabris, 2012*). To investigate the potential regulatory role of the PLC structures in bile duct growth, we established a CCl$_4$-induced mouse model of liver fibrosis. The results showed: (1) using MCNP-Pink to label the portal vein combined with three-dimensional DAB immunohistochemical staining for CK19 to visualize bile duct epithelial cells, EDOF imaging revealed that in control mice, bile duct termini localized adjacent to the portal vein, whereas after 3 or 6 weeks of CCl$_4$ treatment, bile duct termini significantly extended into the hepatic parenchyma by 100–300 μm (*Figure 5C and D*), with a notable increase in bile duct branch area (*Figure 5—figure supplement 1C*). (2) Multiplex TSA 3D imaging demonstrated that under fibrotic conditions, PLC structures became elongated and extended toward the lobular parenchyma. Concurrently, the CD34$^+$Sca-1$^+$ endothelial cell population of PLC showed a significant increase during liver fibrosis progression (*Figure 5—figure supplement 1D*). (3) Further 3D staining confirmed that under fibrotic conditions, bile ducts progressively grew and branching along the expanded PLC structures and infiltrated deeper into the hepatic lobules, forming extensive terminal bile duct branches within the parenchyma. (4) With advancing fibrosis, the extension of CK19$^+$ bile duct termini into the parenchyma was markedly greater after 6 weeks of CCl$_4$ treatment compared to 3 weeks, suggesting a coordinated expansion of bile duct branches alongside PLC structures that correlates with fibrosis severity (*Figure 5E and F*).

Moreover, single-cell transcriptomic re-analysis revealed significant upregulation of bile duct-related genes in the *Cd34$^+$Ly6a$^+$* endothelium of the PLC in fibrotic liver, with notably high expression of *Lgals1* (Galectin-1) and *Hgf* (*Figure 5G*). Previous studies have shown that Galectin-1 is absent in normal liver parenchyma but highly expressed in intrahepatic cholangiocarcinoma (ICC), correlating with tumor dedifferentiation and invasion (*Bacigalupo et al., 2013*; *Shimonishi et al., 2001*). Additionally, hepatocyte growth factor (HGF), particularly in combination with epidermal growth factor (EGF) in 3D cultures, promotes hepatic progenitor cells to form bile duct-polarized cystic structures (*Tanimizu et al., 2007*). Taken together, these results indicate that, in the context of liver fibrosis, CD34$^+$Sca-1$^+$ endothelial cells within the PLC exhibit transcriptional features associated with the upregulation of bile duct-related genes.

Overall, our data demonstrate that the PLC constitutes a vascular microenvironmental structure that can be stably identified in three-dimensional space between the portal vein and the periportal sinusoidal endothelium. Under normal physiological conditions, the PLC shows a high degree of spatial colocalization with bile duct branches and is characterized by enrichment of CD34$^+$Sca-1$^+$ endothelial

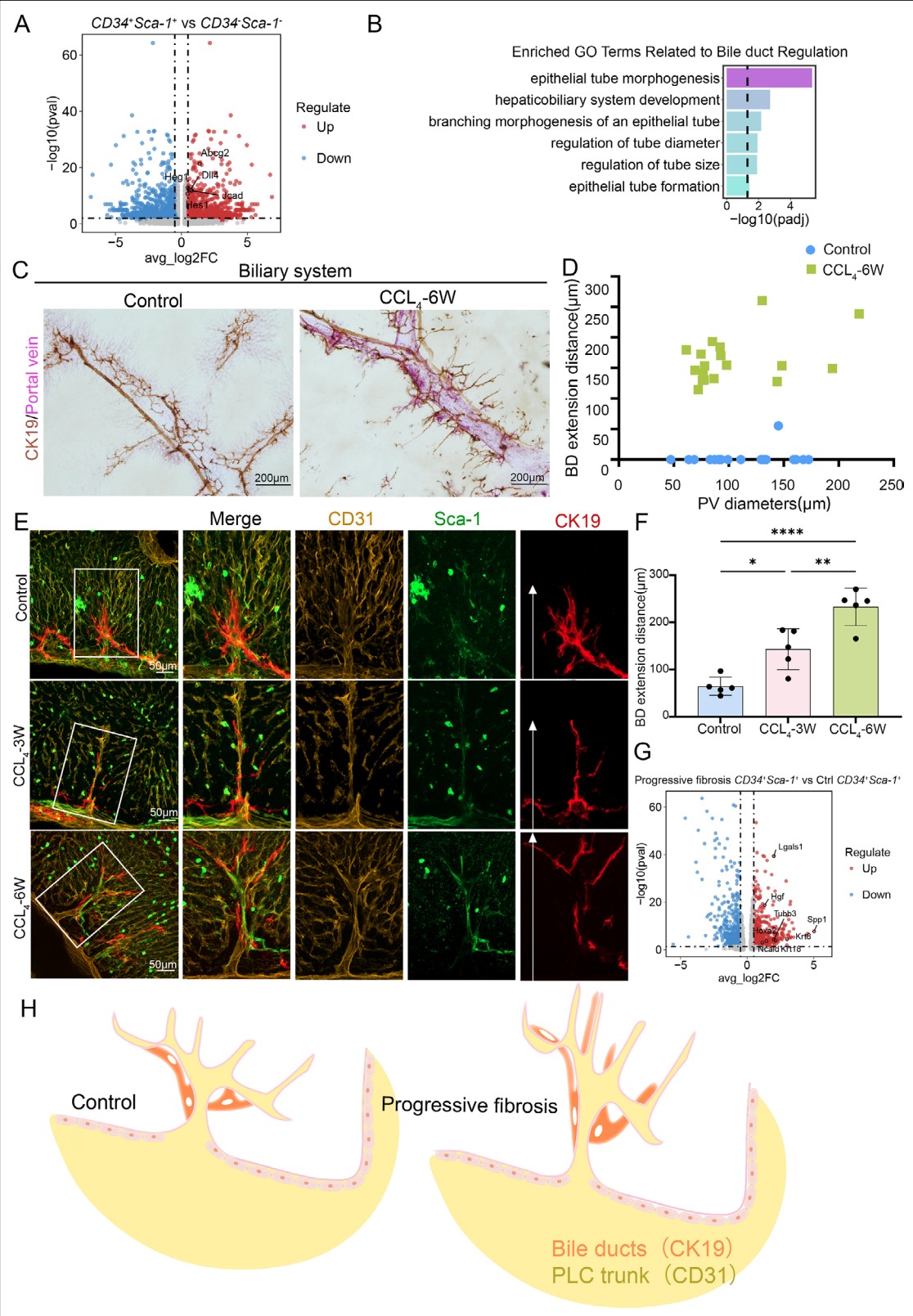

**Figure 5.** CD34+Sca-1+ endothelium in the periportal lamellar complex regulates spatial patterning of intrahepatic bile duct branching during liver fibrosis. (**A**) Volcano plot of differentially expressed bile duct-related genes in CD34+Sca-1+ double-positive cells. The x-axis represents log2 fold change (log2FC), and the y-axis represents -log10 adjusted p value (-log10(p-adjust)). Genes significantly upregulated are located in the upper right quadrant. (**B**) Pathway enrichment analysis of upregulated genes in CD34+Sca-1+ cells showed enrichment in categories such as epithelial morphogenesis and

*Figure 5 continued on next page*

*Figure 5 continued*

branching morphogenesis of epithelial tubes, represented as −log$_{10}$(p value). (**C**) Visualization of portal vein labeled with MCNP-Pink and bile duct epithelial cells stained for CK19 (brown) by 3D DAB immunohistochemistry in control and CCl$_4$ 6-week fibrotic mice. Scale bar: 200 μm. (**D**) Quantification of the distance that bile duct termini extend from portal vein surfaces into hepatic parenchyma in control and CCl$_4$ 6-week mice, presented as mean ± SD (control n=20, CCl$_4$ 6 weeks n=18). (**E**) Multiplex immunofluorescence showing expression and spatial distribution of CD31 (yellow), Sca-1 (green), and CK19 (red) in control and fibrotic models (CCl$_4$ 3 weeks and 6 weeks). White arrows in CK19 single-channel magnified images indicate bile duct termini. Scale bar: 50 μm. (**F**) Quantitative measurement of bile duct termini extension distances along PLC structures into hepatic parenchyma in control, early (CCl$_4$ 3 weeks), and late (CCl$_4$ 6 weeks) fibrosis mice. Data represent mean ± SD (n=5 per group). Statistical significance determined by one-way ANOVA with Tukey's multiple comparisons test; *p<0.05, **p<0.01, ****p<0.0001. (**G**) Volcano plot of differentially expressed bile duct-related genes in CD34$^+$Sca-1$^+$ cells from fibrotic livers compared to controls. Axes as in (**A**). Significantly upregulated genes are in the upper right quadrant. (**H**) Schematic illustration showing spatial localization of CK19 and CD31 within the PLC structure.

The online version of this article includes the following source data and figure supplement(s) for figure 5:

**Source data 1.** Quantification of bile duct termini extension from portal vein surfaces into hepatic parenchyma in control and CCl$_4$ 6-week mice.

**Source data 2.** Quantitative measurement of bile duct termini extension along PLC structures into hepatic parenchyma in control and fibrosis mice at early (CCl$_4$ 3 weeks) and late (CCl$_4$ 6 weeks) stages.

**Figure supplement 1.** Distribution of CD31$^+$, Sca-1$^+$, and CK19$^+$ cells and portal vein–associated bile duct morphology in control and fibrotic mouse livers, related to *Figure 5*.

**Figure supplement 1—source data 1.** Quantification of terminal bile duct area on portal vein surfaces with different diameters in control mice.

cells at its basal region. In fibrotic liver, this structure displays a disease-associated morphological expansion during fibrosis progression (*Figure 5H*).

## CD34$^+$Sca-1$^+$ endothelium in the periportal lamellar complex forms a neurovascular niche regulating hepatic autonomic nerve patterning

In this study, we examined the spatial distribution of hepatic nerves *in mice* using immunofluorescence staining and found that nerve fibers were almost entirely confined to the portal vein region (*Figure 6—figure supplement 1A*). Notably, this distribution pattern differs markedly from that observed in humans—previous studies have shown that, in addition to the portal area, hepatic nerves in humans are also present around the central vein, within the interlobular septa, and in the connective tissue of the parenchyma (*Miller et al., 2021*; *Yi et al., 2010*).

Previous studies have further explained the physiological basis for this difference: even among species that differ in parenchymal sympathetic innervation (i.e., species with or without direct sympathetic input), their sympathetic efferent regulatory functions may remain comparable (*Beckh et al., 1990*). This is because signals released from aminergic and peptidergic nerve terminals can be transmitted to hepatocytes in the form of electrical impulses through intercellular gap junctions (*Hertzberg and Gilula, 1979*; *Jensen et al., 2013*; *Seseke et al., 1992*; *Taher et al., 2017*). However, the scarcity of nerve fibers within the mouse hepatic parenchyma implies that the mechanisms of autonomic regulation of liver function in mice may differ from those in humans, prompting us to further investigate the potential role of PLC endothelial cells in this process.

We further identified that the *CD34$^+$Ly6a$^+$* endothelium of the PLC exhibits prominent neurodevelopmental molecular features. Differential gene expression analysis revealed significant upregulation of multiple genes associated with neurogenesis and axon guidance within this population (*Figure 6A*), including *Nrg1*. Previous studies have shown that NRG-1 is present in intrahepatic nerves and colocalizes with neuronal nitric oxide synthase (nNOS) and vasoactive intestinal peptide (VIP). Exposure to high doses of bisphenol A (BPA) significantly increases the density of NRG-1–immunoreactive fibers in intrahepatic nerves and enhances the colocalization of NRG-1 with VIP, suggesting that NRG-1 in intrahepatic nerves may play a role in BPA-induced liver injury and/or liver regeneration (*Szymańska et al., 2020*); *Adgrg6* (GPR126) might regulate liver innervation through non-myelination-related mechanisms. Indeed, GPR126 has been shown to mediate Schwann cell–dependent and independent processes during peripheral nerve injury and repair, influencing axon guidance, mechanosensing, and ECM remodeling (*Mogha et al., 2016*; *Monk et al., 2009*; *Petersen et al., 2015*). These findings suggest that GPR126 may participate in neural–microenvironment communication within the periportal region (*Monk et al., 2011*). Subsequent GO enrichment analysis of differentially expressed genes (DEGs) in this cell cluster demonstrated significant activation of pathways related to neuronal projection and central nervous system differentiation (*Figure 6B*).

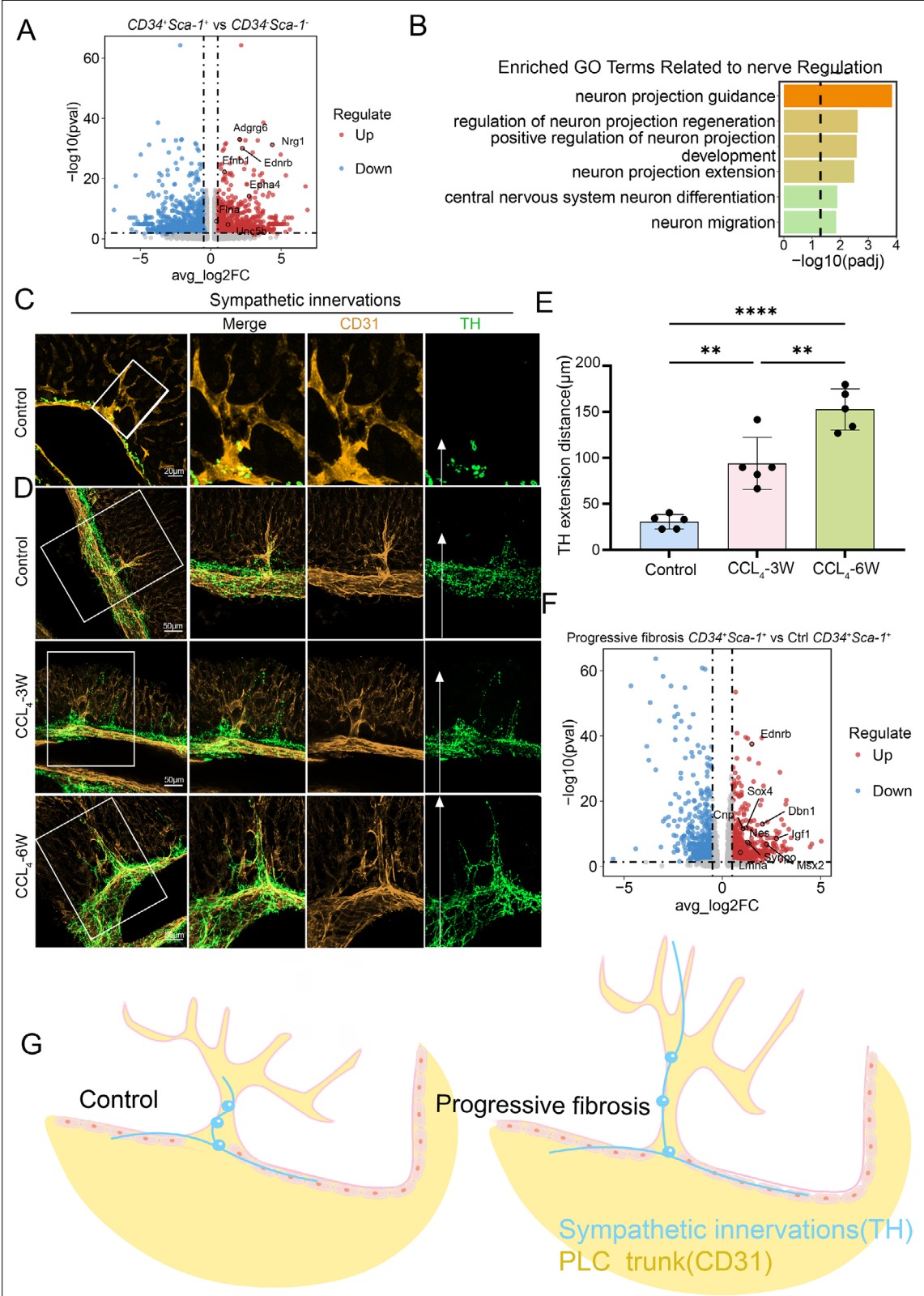

**Figure 6.** CD34+Sca-1+ endothelium in the periportal lamellar complex forms a neurovascular niche regulating hepatic autonomic nerve patterning. (**A**) Differential expression analysis of nerve-related genes in CD34+Sca-1+ endothelial cells. The x-axis indicates log2 fold change (log2FC), and the y-axis represents –log10 adjusted p value (–log10(p-adjust)). Significantly upregulated genes are located in the upper right quadrant. (**B**) Gene Ontology (GO) enrichment analysis of upregulated genes in CD34+Sca-1+ cells, showing enrichment in functional categories such as semaphorin-plexin mediated

*Figure 6 continued on next page*

*Figure 6 continued*

axon guidance, regulation of neuronal projection regeneration, and modulation of postsynaptic neurotransmitter receptor endocytosis. Enrichment significance is indicated by –log10(p-value). (**C**) Multiplex immunofluorescence staining showing tyrosine hydroxylase (TH, green) labeling sympathetic nerves and CD31 (yellow) labeling portal vein endothelium. Scale bar: 20 µm. (**D**) Distribution of CD31 (yellow) and TH (green) expression in control and CCl$_4$-induced liver fibrosis models at week 3 and week 6, visualized by multiplex immunofluorescence. In the green channel images, the white arrows mark the terminal location of TH-positive sympathetic nerve endings. Scale bar: 50 µm. (**E**) Quantification of the distance from sympathetic nerve endings to the hepatic parenchyma along PLC structures in control, early fibrosis (CCl$_4$–3 weeks), and advanced fibrosis (CCl$_4$–6 weeks) mice. Data are presented as mean ± standard deviation (Mean ± SD), n=5 per group. Statistical analysis was performed using one-way ANOVA followed by Tukey's multiple comparison test; *p<0.05, **p<0.01, ****p<0.0001. (**F**) Differential expression analysis of nerve-related genes between CD34$^+$Sca-1$^+$ cells from fibrosis and control livers. The x-axis indicates log2 fold change (log2FC), and the y-axis represents –log10 adjusted p value (–log10(p-adjust)). Significantly upregulated genes are located in the upper right quadrant. (**G**) Schematic diagram illustrating the spatial localization of TH-positive sympathetic nerves and CD31-positive endothelial cells within PLC structures.

The online version of this article includes the following source data and figure supplement(s) for figure 6:

**Source data 1.** Quantification of the distance from sympathetic nerve endings to hepatic parenchyma along PLC structures in control and fibrosis mice (early and advanced stages).

**Figure supplement 1.** Distribution of TH$^+$ sympathetic nerve fibers in mouse and human liver, and their spatial association with hepatic vessels.

**Figure supplement 1—source data 1.** Source data for Figure 6: Re-analysis of dataset from *Pietilä et al., 2025*.

Previous studies have confirmed that the highest density of hepatic parenchymal innervation originates from postganglionic sympathetic neurons located in the celiac ganglion (*Mizuno and Ueno, 2017*). Tyrosine hydroxylase (TH) immunohistochemistry is the preferred method to visualize noradrenergic sympathetic innervation of hepatocytes and vasculature in both rodents and humans (*Adori et al., 2021*). In this study, confocal three-dimensional imaging revealed that TH$^+$ sympathetic nerve fibers form a characteristic 'neuronal bead-chain' network along the PLC structure, with nerve terminals terminating at the PLC base and around vascular bundles, suggesting a potential regulatory role of the PLC in controlling functional zones within the hepatic lobule (*Figure 6C*, *Figure 6—figure supplement 1B and C*).

Further validation using a CCl$_4$-induced mouse liver fibrosis model demonstrated that (1) during fibrosis progression, sympathetic nerve fibers progressively extend along the PLC structure, accompanied by nerve terminal branching and infiltration into the hepatic parenchyma; (2) after 6 weeks of CCl$_4$ treatment, TH$^+$ nerve fibers extended significantly farther into the hepatic parenchyma compared to 3 weeks, indicating a coordinated expansion trend of bile duct branches, PLC structures, and sympathetic nerve networks closely associated with fibrosis severity (*Figure 6D and E*).

Furthermore, single-cell transcriptomic volcano plot re-analysis revealed significant upregulation of neural-related genes within the *CD34$^+$Sca-1$^+$* endothelium of PLC in the fibrosis group, notably including EDNRB (endothelin receptor type B), which plays a critical role in the development of neural crest-derived autonomic nervous system components, including the liver (*Tanimizu et al., 2007*). Nestin, a type VI intermediate filament protein predominantly expressed in the central nervous system, has recently been implicated in tissue homeostasis during wound healing. For example, in traumatic CNS injury, Nestin is induced in reactive astrocytes contributing to glial scar formation. Additionally, Nestin expression correlates with the severity of renal tubulointerstitial fibrosis. In the healthy adult liver, Nestin is nearly absent but is upregulated following acute or chronic hepatic injury, indicating its potential role in liver regeneration (*Figure 6F*; *Ahmed et al., 2013*).

To further validate the reliability of our findings, we incorporated another high-depth single-cell RNA sequencing (scRNA-seq) dataset for cross-analysis and re-exploration (*Pietilä et al., 2025*). This dataset provides a detailed molecular atlas of the major cell types in the adult mouse liver. In their original analysis, Pietilä et al. annotated cluster #e2 as portal vein endothelial cells based on well-established cell-type-specific markers, such as Ly6a/Sca-1 and Cd34 for portal vein endothelium. This annotation closely aligns with the CD34$^+$Sca-1$^+$ endothelial cell population we identified in the PLC region, providing cross-dataset preliminary validation for the existence and core molecular features of this endothelial subpopulation.

Building on this, we performed targeted re-analysis of the reference dataset: by selecting the CD34$^+$Sca-1$^+$ double-positive cell population, we found that their highly expressed gene profile closely matched our own database results, notably including genes related to hematopoietic stem cells (e.g., *ACE*, *Mecom*), bile duct-associated genes (e.g., *DLL4*, *JCAD*), and neural-associated genes

(e.g., *NRG1*, *Adgrg6*), all of which were significantly upregulated (*Figure 6—figure supplement 1D*, *Figure 6—source data 1*).

Meanwhile, GO enrichment analysis of this cell population further confirmed that pathways related to hematopoietic stem cell function regulation (including stem cell development and differentiation), neural development (e.g., neural development), and bile duct-related processes (epithelial tube formation pathway) were all significantly upregulated (*Figure 6—figure supplement 1E*). This result echoes the gene expression characteristics, as it not only strengthens the functional associations between CD34⁺Sca-1⁺ endothelial cells in the PLC region and the hematopoietic, bile duct, and nervous systems, but also provides molecular evidence for their multipotential regulatory properties.

These findings suggest that this specific endothelial subpopulation of PLC may participate in neural branching development and fibrotic microenvironment remodeling, exhibiting multifunctional regulatory potential (*Figure 6G*).

## Discussion

Accurate three-dimensional mapping of the liver is crucial for understanding its function and disease mechanisms. Conventional 2D histology and flow cytometry lose spatial information, particularly in the liver's complex ductal–vascular systems. To address technical challenges in 3D liver imaging—namely, tissue opacity due to heme, lipids, and dense connective tissue, and trade-offs between resolution and field of view—we developed the Liver-CUBIC system. By optimizing $H_2O_2$ bleaching and urea concentration, tissue transparency was significantly improved, while multicolor MCNPs enabled four-channel labeling of portal veins, hepatic arteries, central veins, and bile ducts. This approach allows high-resolution 3D visualization of fine microstructures, including terminal bile duct arborization, portal vein branching, and central vein fenestrations, providing a robust platform for studying liver microcirculation (*Hankeova et al., 2021*; *Peeters et al., 2018*; *Qi et al., 2019*; *Renier et al., 2014*; *Tainaka et al., 2014*).

The labeling strategy relies on perfusing contrast-complementary MCNPs into multiple vascular and ductal compartments. By carefully controlling injection site, pressure, and volume—combined with 15 µL microinjection needles and post-perfusion vessel ligation—we ensured even dye distribution and structural fidelity. High-density image data can be simplified using dual-channel overlays (e.g., portal vein and bile duct), and multiple imaging modalities (e.g., stereomicroscopy, confocal, or extended-depth microscopy) can be chosen depending on the desired resolution or field of view. Moreover, the dyes demonstrate robust fluorescence under laser-scanning microscopy, enabling clear 3D structural imaging.

It should be noted that the current imaging thickness of our method is mostly around 200 µm, and it is not yet possible to achieve whole-liver tissue three-dimensional reconstruction at the level of light-sheet microscopy. However, through MCNP perfusion experiments on intact, unsectioned liver lobes, we have demonstrated dye distribution results covering the entire liver lobe (as shown in *Figure 1—figure supplement 1F*): the images fully present the three-dimensional branching pattern and distribution range of the bile duct system within the liver lobe, clearly showing the continuous network structure from the main trunk to the terminal branches. This confirms that our study can also achieve overall visualization of the vascular system at the whole-organ level.

Although this system has significant advantages, it remains sensitive to anatomical variations of the liver, such as lobe overlap and heterogeneous vascular architecture. At vascular intersections, local perfusion may be uneven or excessively aggregated; therefore, injection strategies and perfusion parameters need to be adjusted according to liver size and vessel status to improve reproducibility and imaging quality. It should be noted that the results obtained with this method are primarily intended to visualize the overall and fine structures of the hepatic vascular and ductal systems, rather than to precisely reflect hemodynamic processes. In the future, we plan to further investigate the diffusion characteristics of the dyes within the liver microcirculation using in vivo perfusion or dynamic fluid models.

Using this system, we identified a previously unrecognized structure located on the adventitia of the main portal vein, which we termed the PLC. The PLC consists of multi-branched extensions radiating from the outer membrane of the main portal vein. Notably, it is absent in terminal vessels and other vascular systems, challenging the conventional 'strict hierarchical bifurcation' model of vascular organization. The PLC is spatially associated with terminal branches of both bile ducts and

hepatic arteries, suggesting possible roles in local microenvironmental regulation through physical or signaling interactions.

Multiplex immunostaining revealed that cholangiocytes extend along ZO-1⁺ bile canalicular networks but display an elongated, immature morphology without forming complete lumens (*Figure 3F*). This observation aligns with previous findings that HNF4α⁺ hepatocytes and CK19⁺ cholangiocytes can jointly form bile-excreting conduits, supporting a guiding role for the PLC in bile duct positioning and morphogenesis (*Tanimizu et al., 2021*).

From a morphological perspective, the PLC exhibits certain features that are similar to those previously described for telocytes (TCs). Telocytes are a class of interstitial cells that have been identified in the liver in recent years by transmission electron microscopy (TEM) and immunohistochemical analyses, and are characterized by CD34 expression and long, slender cellular processes (*Xiao et al., 2013*; *Xu et al., 2019*). However, functional and molecular characterization of TCs remains limited, and their direct relationship with the PLC is yet to be determined.

Single-cell transcriptomic analysis revealed a CD34⁺Sca-1⁺ endothelial cell population enriched within the PLC that expressed hematopoiesis-related genes and stem cell regulatory pathways (*Su et al., 2021*). These results suggest that, beyond structural support, the PLC in the portal region is enriched with perivascular endothelial cell populations exhibiting hematopoiesis-related gene expression features. Supporting this, Sca-1 expression is localized to the portal region during mouse liver development (*Gómez-Salinero et al., 2022*), and Nestin⁺NG2⁺ precursor cells in the fetal liver also concentrate near the portal vein, where they express key HSC-supportive factors such as SCF and CXCL12 (*Ahmed et al., 2013*). Together, these findings associate the PLC with portal vascular regions enriched in hematopoiesis-related cellular and molecular characteristics.

Further analysis revealed upregulation of multiple genes associated with neurodevelopment and axonal guidance within the CD34⁺Sca-1⁺ cluster, accompanied by enrichment of neuronal signaling pathways. Notably, axon guidance-related signaling has also been described in the portal region during liver development, including mesenchyme-mediated guidance mechanisms. Immunostaining further demonstrated TH⁺ sympathetic nerve fibers arranged in a 'beads-on-a-string' pattern surrounding the PLC (*Figure 6*). This spatial configuration resembles previously reported neurovascular structural associations (*Adori et al., 2021*). In addition, sympathetic nerves have been shown to enter the liver along collagen fibers of Glisson's capsule and to run in proximity to hepatic arteries, portal veins, and bile duct epithelium. Taken together, these observations suggest that the PLC may contribute to structural organization within the intrahepatic neurovascular microenvironment.

In CCl₄-induced hepatotoxic fibrosis models, injury and toxic metabolite accumulation are primarily restricted to the centrilobular region, while bile ducts extend linearly toward the central vein (*Kaneko et al., 2015*; *Yang et al., 2021*). We observed dynamic elongation of the PLC in parallel with fibrosis progression, indicating its role as a spatial scaffold for ductal expansion. This suggests potential involvement in ductular reactions and neural remodeling during fibrosis.

In conclusion, this study provides the first systematic characterization of the PLC as a structural and functional interface among hepatic vasculature, bile ducts, and nerves. The PLC functions as a directional scaffold for ductal growth, displays distinct perivascular endothelial transcriptional features in the portal region, and may represent a potential site of neurovascular coupling. These findings provide novel insights into hepatic homeostasis, bile duct development, and fibrotic remodeling, and suggest that the PLC and its CD34⁺Sca-1⁺ endothelial population may represent promising therapeutic targets for diseases such as cholangiocarcinoma and primary sclerosing cholangitis.

## Materials and methods

**Key resources table**

| Reagent type (species) or resource | Designation | Source or reference | Identifiers | Additional information |
| --- | --- | --- | --- | --- |
| Antibody | Anti-CK19 (rabbit monoclonal) | HUABIO | Cat# ET1601-6; RRID:AB_3069617 | 1:3000 |
| Antibody | Anti-CD31 (rabbit monoclonal) | Abcam | Cat# ab182981; RRID:AB_2920883 | 1:2000 |

*Continued on next page*

*Continued*

| Reagent type (species) or resource | Designation | Source or reference | Identifiers | Additional information |
|---|---|---|---|---|
| Antibody | Anti-HNF4A (rabbit monoclonal) | HUABIO | Cat#HA721006; RRID:AB_3072131 | 1:2000 |
| Antibody | Anti-ZO-1 (rabbit polyclonal) | Proteintech | Cat# 21773-1-AP; RRID:AB_10733242 | 1:1000 |
| Antibody | Anti-CD34 (rabbit monoclonal) | HUABIO | Cat# ET1606-11; RRID:AB_2924309 | 1:1000 |
| Antibody | Anti-LY6A (Sca-1) (rabbit monoclonal) | HUABIO | Cat# ET1703-67; RRID:AB_3070422 | 1:500 |
| Antibody | Anti-TH (rabbit monoclonal) | AiFang biological | Cat# AFRM0092 | 1:1000 |
| Antibody | Anti-CD36 (rabbit monoclonal) | HUABIO | Cat# ET1701-24, RRID:AB_3070192 | 1:1000 |
| Antibody | Anti-PDGFR beta (rabbit monoclonal) | HUABIO | Cat# ET1605-20; RRID:AB_3069707 | 1:2000 |
| Antibody | Anti-a-SMA (rabbit monoclonal) | HUABIO | Cat#ET1607-53; RRID:AB_3069772 | 1:5000 |

## Animals

All animal procedures were approved by the Institutional Animal Care and Use Committee of Sichuan University. Eight-week-old male wild-type mice (purchased from Beijing Huafukang Bioscience Co., Ltd.) were used to establish a progressive liver fibrosis model. Mice were intraperitoneally injected with carbon tetrachloride ($CCl_4$, 1 mL/kg body weight, $CCl_4$:olive oil = 1:4, v/v, Damas-beta, CAS:56-23-5) twice a week for 3 or 6 weeks (n=3 per group). Control mice received an equal volume of olive oil injection for 3 or 6 weeks (n=3 per group).

## $CCl_4$-induced liver fibrosis model

Step 1: Preparation of $CCl_4$ solution: Dissolve $CCl_4$ in corn oil at a 1:4 (v/v) ratio under a fume hood and filter through a 0.2 µm filter into a 30 mL vial. Mice were injected intraperitoneally at 1 mL/kg to induce chronic liver injury.

Step 2: Mouse weight and injection volume: Weigh each mouse to calculate the required injection volume.

Step 3: Loading syringe: Draw the calculated $CCl_4$ solution into a syringe and remove air bubbles.

Step 4: Mouse positioning: Place the mouse on a textured surface, lower the head by ~30°, and identify the lower left or right abdominal quadrant.

Step 5: Injection: Insert the needle at a 45° angle (bevel up) approximately 4–5 mm into the abdomen and inject the calculated volume.

Step 6: Post-injection care: Return the mouse to its cage and monitor for any signs of distress. Dispose of syringes and needles safely; a new syringe and needle were used for each mouse.

Step 7: Injection schedule: Administer injections twice weekly for 3 or 6 weeks (n=3 per group). Control mice received equal volumes of corn oil.

Step 8: Tissue collection: 48 h after the last injection, mice were sacrificed, livers were dissected, and samples were processed for histology or molecular analyses.

## Human liver tissue samples

The human liver tissue samples shown in Figure S6 were obtained from adjacent non-tumor liver tissues resected during surgical operations at West China Hospital, Sichuan University. All samples used were anonymized archived tissues, which were applied for scientific research in accordance with institutional ethical guidelines and did not involve any identifiable patient information. After being fixed in 10% neutral formalin for 24 h, the tissues were routinely processed for paraffin embedding (FFPE), and sectioned into 4 µm-thick slices for immunostaining and fluorescence imaging.

## Spectral characteristics of MCNPs

The different types of MCNPs used in this study have distinct spectral characteristics, with specific parameters as follows:

> Both MCNP-Green and MCNP-Yellow are AF488 spectrum-matched types, with an excitation wavelength of 495 nm and an emission wavelength of 519 nm, which are suitable for conventional fluorescence imaging channels.
>
> MCNP-Pink is designed with a far-red spectrum, featuring an excitation wavelength of 561 nm and an emission wavelength of 640 nm. This design can effectively reduce the interference of tissue autofluorescence and improve the signal-to-noise ratio of deep-tissue imaging.
>
> MCNP-Black has no fluorescent properties and only appears black under a bright-field microscope. It is mainly used to form a contrast with fluorescent MCNPs and assist in verifying the integrity of cavity perfusion.

## Four-channel vascular typing labeling and bile duct high-pressure dye perfusion in mice

> Step 1: Mice were euthanized by intraperitoneal (IP) injection of sodium pentobarbital at a dose of 100–150 mg/kg body weight.
>
> Step 2: The chest muscles were incised to expose the heart. A perfusion needle was inserted into the left ventricle, ensuring entry into the cardiac cavity.
>
> Step 3: Sequential perfusion was performed with 40 mL of pre-chilled PBS (5 mL/min) to clear blood, followed by 20 mL of 4% paraformaldehyde (5 mL/min) for in situ fixation.
>
> Step 4: The abdominal skin and muscles were cut to the mid-axillary line to expose the liver. The intestines were gently displaced with sterile swabs to fully expose the common bile duct.
>
> Step 5: According to experimental design, the hepatic artery, bile duct, portal vein, and inferior vena cava were selectively labeled with metallic nanoparticle (MCNP) dyes of different colors.

- Hepatic artery: MCNP-Yellow was injected slowly via the left ventricle using a micro-syringe (25 µL, outer needle diameter 0.31 mm, Shanghai Bolige Industrial Co., Ltd.) until complete arterial filling or resistance occurred.
- Bile duct: After blunt separation of the common bile duct and portal vein, MCNP-Green was retrogradely injected via the extrahepatic bile duct until surface bile ducts were fully filled or resistance was felt. The bile duct was ligated with 5–0 suture.
- Portal vein: MCNP-Pink was injected via the portal vein using a micro-syringe until filling or resistance, followed by ligation.
- Inferior vena cava: MCNP-Black was retrogradely injected via the inferior vena cava, followed by ligation.

> Step 6: The liver was harvested and fixed in 4% paraformaldehyde at 4°C for 48 h, after which tissue clearing procedures were performed.
>
> Step 7: Labeled liver tissues were sectioned to desired thickness and imaged using a high-magnification deep-focus microscope (Keyence), confocal laser scanning microscope (Zeiss LSM980), or THUNDER high-resolution imaging system (Leica).
>
> Note: The color assignment of MCNP dyes is flexible and does not represent a fixed correspondence to specific vascular or ductal structures; different colors were selected solely for visualization and presentation purposes to facilitate multichannel imaging.

## Bile duct high-pressure dye perfusion

Follow steps 1–4 as above. Then, using a micro-syringe, retrograde catheterization was performed via the extrahepatic bile duct with secure positioning at the hepatic hilar bile duct. A large volume of MCNP-Green was rapidly injected until the entire bile duct system and peripheral branches were filled, evident from surface dye visualization. Dye volume was adjusted to liver size, typically 25–40 µL per mouse.

## Traditional CUBIC reagent preparation and tissue clearing

Based on Hiroki R. Ueda's protocol (*Susaki et al., 2014*):

Step 1: Prepare ScaleCUBIC-1 (25% CUBIC-I): 25 wt% urea, 25 wt% N,N,N',N'-tetrakis(2-hydroxypropyl)ethylenediamine, and 15 wt% Triton X-100. Prepare ScaleCUBIC-2 (25% CUBIC-II): 50 wt% sucrose, 25 wt% urea, 10 wt% triethanolamine, and 0.1% (v/v) Triton X-100.
Step 2: Fixed liver lobes were immersed in 20 g of CUBIC-I at 37°C with gentle shaking for 3 days, then replaced with fresh CUBIC-I for another 2 days. Samples were rinsed with PBS and immersed in CUBIC-II for 3 days.

## Liver-CUBIC (40%CUBIC+H$_2$O$_2$) reagent preparation and tissue clearing

Step 1: Prepare 40%CUBIC-I: 36 wt% urea, 25 wt% N,N,N',N'-tetrakis(2-hydroxypropyl)ethyl-enediamine, and 15 wt% Triton X-100. Prepare 40%CUBIC-II: 50 wt% sucrose, 35 wt% urea, 10 wt% triethanolamine, and 0.1% Triton X-100. Bleaching solution: 4.5% hydrogen peroxide and 24 mM sodium hydroxide.
Step 2: We subjected all hepatic lobes of the whole liver (including the left, right, caudate, and quadrate lobes) to the Liver-CUBIC aqueous clearing protocol to ensure uniform visualization of MCNP fluorescence and immunolabeling signals throughout the three-dimensional reconstruction of the entire liver. The liver lobes were immersed in IRISKit HyperView Quench Buffer (Cat. #MH010301) at room temperature (RT) for 24 h, with the solution refreshed every 12 h. The samples were then washed with PBS and incubated in 20 g of 40%CUBIC-I solution at 37°C for 1 day, washed again, and subsequently incubated in 40%CUBIC-II solution at 37°C for another 1 day.

## Transmission quantification

Transmittance was measured using a microplate reader (BioTek, USA) on 1-mm-thick liver sections.

Step 1: 1 mm liver slices were cut into 5 mm × 5 mm squares and treated with various clearing protocols, then immersed in CUBIC-II. Five replicates per group were placed into 96-well plates.
Step 2: Optical transmittance was recorded from 400 nm to 900 nm, generating transmittance spectra for each sample.

## TEM of MCNPs

Prior to imaging, MCNP dye solutions were centrifuged at 14,000×*g* for 10 min at 4°C to remove aggregates and impurities. The supernatant was diluted 50-fold, and 3–4 µL of the diluted sample was applied onto freshly glow-discharged Quantifoil R1.2/1.3 copper grids (Electron Microscopy Sciences, 300 mesh). After a 30 s incubation to allow particle adsorption, excess liquid was gently blotted with filter paper and the grid was air-dried at RT. Samples were negatively stained with 1% uranyl acetate for 30 s and air-dried again before imaging.

Negative-stain TEM images were acquired using a 120 kV JEOL JEM-1400 transmission electron microscope (JEOL Ltd., Japan) equipped with a CCD camera. Data collection was performed using standard imaging conditions.

## 3D DAB staining and imaging

Step 1: Liver tissues were fixed in 4% paraformaldehyde at 4°C for 24 h, sectioned to 200 µm using a vibratome (Leica VT1200, Germany), and washed three times in distilled water (10 min each), DAB staining was performed using free-floating sections. The sections were processed while floating in solution throughout the entire staining procedure.
Step 2: Sections were delipidated in acetone: chloroform (1:1, v/v) at RT for 3 h.
Step 3: Dehydrated in graded ethanol (75%, 85%, 95%, 100%, 100%, 95%, 85%, 75%), 30 min each, followed by three 10 min water washes.
Step 4: Quenching was performed with quench reagent (including H$_2$O$_2$, IRISKit HyperView Quench Buffer) for 20 min in a LUMINIRIS quencher, followed by three water washes. Sections

were treated with 2% EDTA antigen retrieval buffer (in distilled water) at 95°C for 35 min, then washed with PBS +0.3% Triton X-100 three times (10 min each).

Step 5: Incubated with primary antibody (rabbit anti-CK19, 1:2000, Huaan Biotechnology ET1601-6) at 4°C for 24 h, followed by three PBS washes.

Step 6: Incubated with polymerized HRP-conjugated secondary antibody at 4°C for 24 h, followed by PBS +0.3% Triton washes.

Step 7: Stained with DAB (Zhongshan Jinqiao ZLI-9017) at RT for 10 min, followed by three PBS washes.

Step 8: Sequentially cleared with 40% CUBIC-I at 37°C for 30 min, washed, and incubated in 40% CUBIC-II at 37°C for 30 min.

Step 9: 3D imaging was performed using a high-depth focus microscope.

## 3D TSA multiplex immunofluorescence staining

Performed using IRISKit HyperView multiplex immunostaining kit (#MH010101).

Steps 1–4: Same as 3D DAB protocol.

Step 5: Incubated with primary antibodies (anti-CK19, CD31, HNF4α, ZO-1, CD34, Sca-1, and TH) overnight at 4°C.

Step 6: Incubated with polymerized HRP-conjugated secondary antibody overnight at 4°C.

Step 7: Sequentially visualized each antibody with specific TSA fluorophores: Cyclic-480 (EX 465–495 nm, EM 512–558 nm), Cyclic-550 (EX 540–560 nm, EM 575–595 nm), and Cyclic-630 (EX 620–640 nm, EM >665 nm).

Step 8: After each round, antibodies were stripped using IRISKit HyperView advanced antibody removal kit.

Step 9: Cleared tissue was imaged by confocal laser scanning microscopy (Zeiss LSM980).

## Imaging equipment and parameter settings

In this study, three types of microscopes were employed for imaging analysis. The specific models and parameter settings are described as follows:

## Ultra-Depth Microscope (VHX-6000, Keyence)

This microscope was equipped with a VH-Z100R objective lens, offering a magnification range of 100×–1000× and a typical resolution of 1 μm. The illumination mode was set to coaxial reflected light, and the platform's transmitted illumination was turned 'ON' to ensure uniform brightness across the tissue samples, thereby facilitating clear visualization of structural details.

## Zeiss Confocal Microscope (LSM980)

Depending on imaging requirements, either a 20× or 40× objective was used, with the image size set to 1024×1024 pixels. Three fluorescence detection channels were configured as follows:

- Channel 1 (far-red): 639 nm laser excitation; emission was collected at 673–758 nm.
- Channel 2 (orange-red): 561 nm laser excitation; emission was collected at 547–637 nm.
- Channel 3 (green): 488 nm laser excitation; emission was collected at **490–529** nm.

## Leica THUNDER Imager 3D Tissue Microscope

This microscope was configured with two fluorescence detection channels:

Channel 1: FITC channel ($\lambda_{ex} = 488$ nm; $\lambda_{em} \approx 520$ nm);

Channel 2: Orange-red channel ($\lambda_{ex} = 561$ nm; $\lambda_{em} \approx 640$ nm).

To ensure signal specificity, a dedicated set of filters matched to the spectral characteristics of both channels was applied, effectively minimizing crosstalk between different fluorescent signals.

## 3D image quantification

ImageJ software was used to analyze 3D stained images (*Figures 5E and 6E*), and the ultra-depth-of-field 3D analysis module was used to analyze 3D DAB images (*Figure 5D*). The specific steps are as follows:

> *Figure 5D*: DAB-stained 3D images from the control group and the CCl$_4$ 6 weeks (CCl$_4$–6 W) group were analyzed. For each group, 20 terminal bile duct branch nodes were randomly selected, and the actual path distance along the branch to the nearest portal vein surface was measured. All measurements were plotted as scatter plots to reflect the spatial extension of bile ducts relative to the portal vein under different conditions.
> *Figure 5F*: TSA 3D multiplex-stained images from the control group, CCl$_4$ 3 weeks (CCl$_4$–3W), and CCl$_4$ 6 weeks (CCl$_4$–6W) groups were analyzed. For each group, five terminal bile duct branch nodes were randomly selected, and the actual path distance along the branch to the nearest portal vein surface was measured. Measurements were plotted as scatter plots to illustrate bile duct spatial extension.
> *Figure 6E*: TSA 3D multiplex-stained images from the control, CCl$_4$–3W, and CCl$_4$–6W groups were analyzed. For each group, five terminal nerve branch nodes were randomly selected, and the actual path distance along the branch to the nearest portal vein surface was measured. Scatter plots were generated to depict the spatial distribution of nerves under different treatment conditions.

## Single-cell data mining

We re-analyzed published single-cell transcriptomic datasets of liver endothelial cells (ECs) sorted by FACS from three normal mouse livers (*Su et al., 2021*). In this study, we downloaded single-cell RNA sequencing (scRNA-seq) data published by Su et al. from a public database and preprocessed the data according to the filtering criteria reported in the original study, retaining only GFP-positive cells.

Subsequently, standard analysis procedures were performed using Seurat (v5.0.3), including quality control, normalization, feature selection, dimensionality reduction, and clustering. Cell populations were annotated based on marker genes reported in the literature, and UMAP was used to visualize cell distribution. We further quantified the abundance of different endothelial cell (EC) subpopulations. After extracting the Seurat object for ECs, clustering results were corrected and re-clustered. Differential expression analysis was performed using the FindMarkers function. During this process, we selected the intersection of highly variable genes and differentially expressed genes in ECs as a candidate gene set and generated a heatmap showing the top 20 differentially expressed genes from portal vein ECs to periportal LSECs. Within this intersecting gene set, Cd34 and Sca-1 were identified. We then extracted the Cd34$^+$Sca-1$^+$ double-positive subpopulation from both portal vein and periportal vein clusters and continued analysis and differential gene screening according to the Seurat workflow.

Finally, the differentially expressed genes (p_val <0.05 and avg_log2FC >0.3) were subjected to GO enrichment analysis with ont = 'BP'. Enrichment bar plots and volcano plots were generated using ggplot2 (v3.5.1) to illustrate key differentially expressed genes and their functional enrichment. The raw data for this re-analysis, including preprocessed expression matrices, clustering results, and differential expression gene lists, are provided in .

To validate these results, we downloaded data from another public dataset containing cell populations similar to the PLC in our study. After extracting the target population (e2), cells were divided into Cd34- and Sca-1-positive and -negative groups, and the previous analyses—including differential expression and enrichment analyses—were repeated for validation (*Pietilä et al., 2025*). The raw data for this re-analysis, including preprocessed expression matrices, clustering results, and differential expression gene lists, are provided in *Figure 6—figure supplement 1—source data 1*.

## Statistics and reproducibility

Statistical analyses were performed using Prism 8.0 (GraphPad Software, Inc). p-values are indicated in figure legends. Data are presented as mean ± SD unless otherwise stated. Two-group comparisons were made using Student's *t*-test. One-way ANOVA followed by Tukey's multiple comparisons

test was used for multi-group analysis. Statistical significance was defined as \*\*\*$p<0.001$, \*\*$p<0.01$, \*$p<0.05$.

## Acknowledgements

This study was supported by multiple funding sources, including the National Natural Science Foundation of China (grant no. 82270542) and the Natural Science Foundation of Sichuan Province (grant no. 2023NSFSC0665).

## Additional information

### Competing interests

Qin Chen: is affiliated with Chengdu Minghong Tiancheng Technology Co., Ltd. The author has no other competing interests to declare. The other authors declare that no competing interests exist.

### Funding

| Funder | Grant reference number | Author |
|---|---|---|
| National Natural Science Foundation of China | 82270542 | Chengjian Zhao |
| Natural Science Foundation of Sichuan Province | 2023NSFSC0665 | Chengjian Zhao |

The funders had no role in study design, data collection and interpretation, or the decision to submit the work for publication.

### Author contributions

Tongtong Xu, Conceptualization, Data curation, Formal analysis, Validation, Investigation, Visualization, Methodology, Writing - original draft; Fujun Cao, Data curation, Formal analysis, Methodology; Ruihan Zhou, Validation, Methodology; Qin Chen, Data curation; Jian Zhong, Formal analysis; Yulin Wang, Banglei Yin, Investigation; Chaoxin Xiao, Methodology; Chong Chen, Validation, Writing – review and editing; Chengjian Zhao, Conceptualization, Supervision, Funding acquisition, Project administration, Writing – review and editing

### Author ORCIDs

Chong Chen ⓘ https://orcid.org/0000-0002-6787-0495
Chengjian Zhao ⓘ https://orcid.org/0000-0003-0624-5263

### Ethics

All animal procedures described in this study were conducted under a protocol reviewed and approved by the Experimental Animal Ethics Committee of the State Key Laboratory of Biotherapy, Sichuan University (approval no. 20220531067; approved on September 10, 2022). The approved project was entitled A Novel 3D Visualization Method in Mice Identifies the Periportal Lamellar Complex (PLC) as a Key Regulator of Hepatic Ductal and Neuronal Branching Morphogenesis. SPF-grade C57BL/6 mice (n = 80) were used in this study and were purchased from Chengdu Dashuo Laboratory Animal Co., Ltd. All surgery was performed under sodium pentobarbital anesthesia, and every effort was made to minimize suffering.

Reviewer #1 (Public review): https://doi.org/10.7554/eLife.108669.5.sa1
Reviewer #3 (Public review): https://doi.org/10.7554/eLife.108669.5.sa2
Author response https://doi.org/10.7554/eLife.108669.5.sa3

# Additional files

## Supplementary files
MDAR checklist

## Data availability
All source data generated during the re-analysis of the aforementioned data in this study, including preprocessed expression matrices, cell clustering results, differential expression gene lists, and functional enrichment analysis results, have been organized and deposited in Figure 4-source data 1 and Figure 6-figure supplement 1-source data 1 of the manuscript, and are accessible alongside the manuscript and its supplementary materials.

The following previously published datasets were used:

| Author(s) | Year | Dataset title | Dataset URL | Database and Identifier |
|---|---|---|---|---|
| Pietilä R, Genové G, Mocci G, Miao Y, Liu J, Leptidis S, Del Gaudio F, Uhrbom M, Vázquez-Liébanas E, Gustafsson S, Buyandelger B, Raschperger E, Björkegren JLM, Hansson EM, Gaengel K, Mäe MA, Jeansson M, Vanlandewijck M, He L, Strell C, Muhl L | 2025 | A comprehensive molecular atlas of mesenchymal cell types in the mouse liver | https://www.ncbi.nlm.nih.gov/geo/query/acc.cgi?acc=GSE297062 | NCBI Gene Expression Omnibus, GSE297062 |
| Su T, Yang Y, Lai S, Jeong J, Jung Y, McConnell M, Utsumi T, Iwakiri Y | 2021 | Single-cell RNA sequencing reveals transcriptome alterations of liver sinusoidal endothelial cells in different zonal microenvironments of cirrhotic livers | https://www.ncbi.nlm.nih.gov/geo/query/acc.cgi?acc=GSE147581 | NCBI Gene Expression Omnibus, GSE147581 |

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
