## [Editor Report · eLife Assessment]

This study presents an **important** methodological advance-Liver-CUBIC combined with multicolor metallic nanoparticle perfusion-that enables high-resolution 3D visualization of the liver's complex multi-ductal architecture. The identification of the periportal lamellar complex (PLC) as a novel perivascular structure with distinct cellular composition and low-permeability characteristics is **convincing**, supported by rigorous imaging data. The observed scaffolding role during fibrosis offers intriguing biological insights, though the functional claims would benefit from direct experimental validation.

---

## [Referee Report · Reviewer #1 (Public review)]

[Editors' note: this version has been assessed by the Reviewing Editor without further input from the original reviewers. The authors have addressed the minor comments raised in the previous round of review.]

Summary:

In this manuscript, Chengjian Zhao et al. focused on the interactions between vascular, biliary, and neural networks in the liver microenvironment, addressing the critical bottleneck that the lack of high-resolution 3D visualization has hindered understanding of these interactions in liver disease.

Strengths:

This study developed a high-resolution multiplex 3D imaging method that integrates multicolor metallic compound nanoparticle (MCNP) perfusion with optimized CUBIC tissue clearing. This method enables the simultaneous 3D visualization of spatial networks of the portal vein, hepatic artery, bile ducts, and central vein in the mouse liver. The authors reported a perivascular structure termed the Periportal Lamellar Complex (PLC), which is identified along the portal vein axis. This study clarifies that the PLC comprises CD34⁺Sca-1⁺ dual-positive endothelial cells with a distinct gene expression profile, and reveals its colocalization with terminal bile duct branches and sympathetic nerve fibers under physiological conditions.

Comments on revisions:

The authors very nicely addressed all concerns from this reviewer. There are no further concerns and comments.

---

## [Referee Report · Reviewer #3 (Public review)]

Xu, Cao and colleagues aimed to overcome the obstacles of high-resolution imaging of intact liver tissue. They report successful modification of the existing CUBIC protocol into Liver-CUBIC, a high-resolution multiplex 3D imaging method that integrates multicolor metallic compound nanoparticle (MCNP) perfusion with optimized liver tissue clearing, significantly reducing clearing time and enabling simultaneous 3D visualization of the portal vein, hepatic artery, bile ducts, and central vein spatial networks in the mouse liver. Using this novel platform, the researchers describe a previously unrecognized perivascular structure they termed Periportal Lamellar Complex (PLC), regularly distributed along the adult liver portal veins.

Using available scRNAseq data, the authors assessed the CD34⁺Sca-1⁺ cells' expression profile, highlighting mRNA presence of genes linked to neurodevelopment, bile acid transport, and hematopoietic niche potential. Different aspects of this analysis were then addressed by protein staining of selected marker proteins in the mouse liver tissue. Next, the authors addressed how the PLC and biliary system react to CCL4-induced liver fibrosis, implying PLC dynamically extends, acting as a scaffold that guides the migration and expansion of terminal bile ducts and sympathetic nerve fibers into the hepatic parenchyma upon injury.

The work clearly demonstrates the usefulness of the Liver-CUBIC technique and the improvement of both resolution and complexity of the information, gained by simultaneous visualization of multiple vascular and biliary systems of the liver. The identification of PLC and the interpretation of its function represent an intriguing set of observations that will surely attract the attention of liver biologists as well as hepatologists. The importance of the CD34+/Sca1+ endothelial cell population and claims based on transcriptomic re-analysis require future assessment by functional experimental approaches to decipher the functional molecules involved in PLC formation, maintenance, and the involvement in injury response before establishing their role in biliary, arterial, and neural liver systems.

Strengths:

The authors clearly demonstrate an improved technique tailored to the visualization of the liver vasulo-biliary architecture in unprecedented resolution.

This work proposes a new morphological feature of adult liver facilitating interaction between the portal vein, hepatic arteries, biliary tree, and intrahepatic innervation, centered at previously underappreciated protrusions of the portal veins - PLCs.

Weaknesses:

The importance of CD34+Sca1+ endothelial cell sub-population for PLC formation and function was not tested and warrants further validation.

---

## [Author Response]

The following is the authors’ response to the previous reviews

**Public Reviews:**

**Reviewer #1 (Public review):**
Summary:In this manuscript, Chengjian Zhao et al. focused on the interactions between vascular, biliary, and neural networks in the liver microenvironment, addressing the critical bottleneck that the lack of high-resolution 3D visualization has hindered understanding of these interactions in liver disease.Strengths:This study developed a high-resolution multiplex 3D imaging method that integrates multicolor metallic compound nanoparticle (MCNP) perfusion with optimized CUBIC tissue clearing. This method enables the simultaneous 3D visualization of spatial networks of the portal vein, hepatic artery, bile ducts, and central vein in the mouse liver. The authors reported a perivascular structure termed the Periportal Lamellar Complex (PLC), which is identified along the portal vein axis. This study clarifies that the PLC comprises CD34⁺Sca-1⁺ dual-positive endothelial cells with a distinct gene expression profile, and reveals its colocalization with terminal bile duct branches and sympathetic nerve fibers under physiological conditions.Comments on revisions:The authors very nicely addressed all concerns from this reviewer. There are no further concerns and comments.

We thank the reviewer for the positive evaluation and helpful feedback.

**Reviewer #3 (Public review):**
Xu, Cao and colleagues aimed to overcome the obstacles of high-resolution imaging of intact liver tissue. They report successful modification of the existing CUBIC protocol into Liver-CUBIC, a high-resolution multiplex 3D imaging method that integrates multicolor metallic compound nanoparticle (MCNP) perfusion with optimized liver tissue clearing, significantly reducing clearing time and enabling simultaneous 3D visualization of the portal vein, hepatic artery, bile ducts, and central vein spatial networks in the mouse liver. Using this novel platform, the researchers describe a previously unrecognized perivascular structure they termed Periportal Lamellar Complex (PLC), regularly distributed along the adult liver portal veins.Using available scRNAseq data, the authors assessed the CD34^+^/Sca-1^+^ cells' expression profile, highlighting mRNA presence of genes linked to neurodevelopment, bile acid transport, and hematopoietic niche potential. Different aspects of this analysis were then addressed by protein staining of selected marker proteins in the mouse liver tissue. Next, the authors addressed how the PLC and biliary system react to CCL4-induced liver fibrosis, implying PLC dynamically extends, acting as a scaffold that guides the migration and expansion of terminal bile ducts and sympathetic nerve fibers into the hepatic parenchyma upon injury.The work clearly demonstrates the usefulness of the Liver-CUBIC technique and the improvement of both resolution and complexity of the information, gained by simultaneous visualization of multiple vascular and biliary systems of the liver. The identification of PLC and the interpretation of its function represent an intriguing set of observations that will surely attract the attention of liver biologists as well as hepatologists. The importance of the CD34+/Sca1+ endothelial cell population and claims based on transcriptomic re-analysis require future assessment by functional experimental approaches to decipher the functional molecules involved in PLC formation, maintenance, and the involvement in injury response before establishing their role in biliary, arterial, and neural liver systems.Strengths:The authors clearly demonstrate an improved technique tailored to the visualization of the liver vasulo-biliary architecture in unprecedented resolution.This work proposes a new morphological feature of adult liver facilitating interaction between the portal vein, hepatic arteries, biliary tree, and intrahepatic innervation, centered at previously underappreciated protrusions of the portal veins - PLCs.Weaknesses:The importance of CD34+Sca1+ endothelial cell sub-population for PLC formation and function was not tested and warrants further validation.

We thank the reviewer for the valuable comment regarding the potential role of the CD34^+^/Sca-1^+^ endothelial cell sub-population in PLC function.

We agree that direct functional validation would be a crucial next step to confirm the contribution of this specific sub-population to PLC formation and function. The focus of the present study remains on the spatial localization and reproducible characterization of PLC structures based on 3D imaging, as well as the relevant transcriptional features revealed by single-cell analysis.

To avoid overinterpretation, we have revised the Discussion section accordingly, providing a more focused and cautious description of the related findings.

Comments on revisions:I appreciate the author's effort to revise the text so it more rigorously adheres to the presented evidence. Following a thorough read of the revised text, a few remaining minor issues were identified in the Discussion.(1) From where comes the hard evidence for PLC being the stem cell niche in the following sentence?for the two following statements:This suggests that the PLC may not only provide structural support but also serve as a perivascular stem cell niche specific to the portal region, potentially involved in hematopoiesis and tissue regeneration.The PLC serves as a directional scaffold for ductal growth, a specialized stem cell niche, and a potential site of neurovascular coupling.

We thank the reviewer for this important comment. We agree that the term “stem cell niche” may imply functional evidence for direct stem cell regulation, which was not demonstrated in this study. Our conclusions were based on the spatial enrichment and transcriptional features of CD34^+^/Sca-1^+^ endothelial populations expressing hematopoiesis-related genes in the portal region.

To avoid overinterpretation, we have revised the sentence to remove the term “stem cell niche” and instead describe the PLC as being enriched in perivascular endothelial cell populations with hematopoiesis-related gene expression features. The revised text now reads:

“These results suggest that, beyond structural support, the PLC in the portal region is enriched with perivascular endothelial cell populations exhibiting hematopoiesis-related gene expression features.”

We have also modified the corresponding statement later in the Discussion. It now reads:

“The PLC serves as a directional scaffold for ductal growth, displays distinct perivascular endothelial transcriptional features in the portal region, and may represent a potential site of neurovascular coupling.”

We believe this wording more accurately reflects the descriptive and transcriptomic nature of our data without implying functional niche activity.

(2) In the following paragraph, I lack references to the previously published evidence of liver innervation guidance mechanisms, such as the mesenchyme-mediated guidance (CD31- population) Gannoun et al., 2023 https://doi.org/10.1242/dev.201642, an important context for your finding.Further analysis showed significant upregulation of genes involved in neurodevelopment and axonal guidance in the CD34^+^/Sca-1^+^ cluster, along with activation of neuronal signaling pathways. Immunostaining confirmed the presence of TH^+^ sympathetic nerve fibers wrapping around the PLC in a "beads-on-a-string" pattern (Fig. 6), consistent with a classic neurovascular unit(Adori et al., 2021). Previous studies have shown that sympathetic nerves enter the liver along collagen fibers of Glisson's capsule and interact with hepatic arteries, portal veins, and bile duct epithelium, supporting the PLC as a scaffold for intrahepatic neurovascular integration.

We thank the reviewer for highlighting the importance of previously published evidence regarding liver innervation guidance mechanisms. We agree that these studies provide important context for interpreting the neurodevelopmental and axon guidance–related transcriptional signatures observed in our dataset. Accordingly, we have revised the Discussion section to incorporate reference to mesenchyme-mediated axon guidance mechanisms in the portal region during liver development (Gannoun et al., 2023). This addition better situates our findings within the existing literature.

(3) Several sentences have issues with a lack of space between words.

We have carefully re-examined the entire manuscript for spacing and formatting inconsistencies and corrected minor typographical issues to ensure uniform formatting throughout the text.